# SADA: Stability-guided Adaptive Diffusion Acceleration

Ting Jiang [* 1]   Yixiao Wang [* 2]   Hancheng Ye [* 1]   Zishan Shao [1 2]   Jingwei Sun [1]   Jingyang Zhang [1]   Zekai Chen [1]
Jianyi Zhang [1]   Yiran Chen [1]   Hai Li [1]

## Abstract

Diffusion models have achieved remarkable success in generative tasks but suffer from high computational costs due to their iterative sampling process and quadratic-attention costs. Existing training-free acceleration strategies that reduce per-step computation cost, while effectively reducing sampling time, demonstrate low faithfulness compared to the original baseline. We hypothesize that this fidelity gap arises because (a) different prompts correspond to varying denoising trajectory, and (b) such methods do not consider the underlying ODE formulation and its numerical solution. In this paper, we propose **Stability-guided Adaptive Diffusion Acceleration (SADA)**, a novel paradigm that unifies step-wise and token-wise sparsity decisions via a single stability criterion to accelerate sampling of ODE-based generative models (Diffusion and Flow-matching). For (a), SADA adaptively allocates sparsity based on the sampling trajectory. For (b), SADA introduces principled approximation schemes that leverage the precise gradient information from the numerical ODE solver. Comprehensive evaluations on SD-2, SDXL, and Flux using both EDM and DPM++ solvers reveal consistent $\geq 1.8\times$ speedups with minimal fidelity degradation (LPIPS $\leq 0.10$ and FID $\leq 4.5$) compared to unmodified baselines, significantly outperforming prior methods. Moreover, SADA adapts seamlessly to other pipelines and modalities: It accelerates ControlNet without any modifications and speeds up MusicLDM by $1.8\times$ with $\sim 0.01$ spectrogram LPIPS. Our code is available at: https://github.com/Ting-Justin-Jiang/sada-icml.

*Equal contribution [1]Department of Electrical and Computer Engineering, Duke University, Durham, U.S.A [2]Department of Statistical Science, Duke University, Durham, U.S.A. Correspondence to: Ting Jiang <justin.jiang@duke.edu>.

*Proceedings of the $42^{nd}$ International Conference on Machine Learning*, Vancouver, Canada. PMLR 267, 2025. Copyright 2025 by the author(s).

## 1. Introduction

Recent advancements in diffusion models have set new benchmarks across various tasks, including image, video, text, and audio generation (Sohl-Dickstein et al., 2015; Song et al., a), significantly enhancing productivity and creativity. However, the deployment of these models at high resolutions faces two fundamental efficiency bottlenecks: (i) the iterative nature of the denoising process and (ii) the quadratic complexity of attention mechanisms. To address these challenges, existing training-free acceleration methods have primarily focused on two corresponding fronts: (i) reducing the number of inference steps (Song et al., a; Lu et al., 2022b; Karras et al., 2022), (ii) reducing the computational cost per step (Bolya & Hoffman, 2023; Ma et al., 2024b; Zhao et al., 2024; Wang et al., 2024; Zou et al., 2024; Ye et al., 2024).

Sampling with generative models can be cast as transporting between two distributions through a reverse ordinary differential equation (ODE) (Song & Ermon, 2019; Song et al., b; Lipman et al., 2023; Liu et al., 2022). Numerical ODE solvers (Karras et al., 2022; Lu et al., 2022a;b) that significantly reduce the number of inference steps while preserving sample fidelity have become a fundamental component for practical workflows. Orthogonal to these advanced schedulers, recent works that reduce per-step computational cost in category (ii) exploit the sparsity empirically observed in pretrained architectures (Rombach et al., 2022; Chen et al., 2024b; Podell et al.; Esser et al., 2024) during the sampling procedure, either on a token-wise (Bolya & Hoffman, 2023; Kim et al., 2024; Zou et al., 2024) or step-wise (Ma et al., 2024b; Zhao et al., 2024; Ye et al., 2024) granularity.

While effectively reducing sampling latency, these architectural strategies in category (ii) demonstrate low faithfulness compared to original samples, as measured in LPIPS (Zhang et al., 2018) and FID (Heusel et al., 2017). We hypothesize that this fidelity gap arises because: (a) Fixed (Ma et al., 2024b) or pre-searched (Yuan et al., 2024) sparsity patterns cannot adapt to the variability of each prompt's denoising trajectory, and (b) Such methods do not explicitly leverage the underlying ODE formulation of the denoising process, nor interplay with the specific ODE-solver used.

Motivated by these observations, we introduce *Stability-*

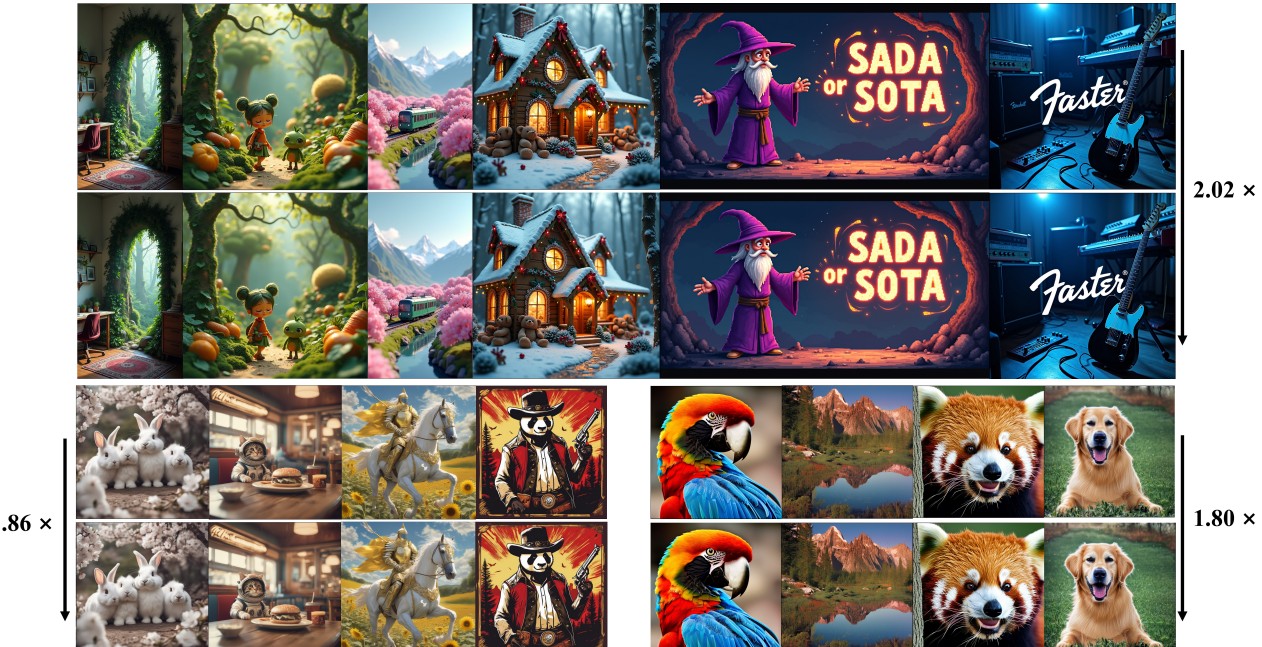

*Figure 1.* Accelerating {`Flux`, `SDXL`, `SD-2`} by {2.02×, 1.86×, 1.80×} with Stability-guided Adaptive Diffusion Acceleration with 50 {Diffusion, Flow-matching} inference steps.

*guided Adaptive Diffusion Acceleration* (**SADA**), a training-free framework that dynamically exploits both step-wise and token-wise sparsity via a unified stability criterion. SADA addresses (a) by adaptively allocating computation along the denoising trajectory (See Section 3.3), and (b) by employing approximation schemes that leverage precise trajectory gradient information from the chosen ODE solver (See Section 3.4).

Specifically, leveraging the second-order difference of the precise gradient $y_t = \frac{\mathrm{d}x_t}{\mathrm{d}t}$ according to its definition (Song et al., a; Lipman et al., 2023), SADA makes better decisions on sparsity allocation and principled approximation. For (a), we propose the stability criterion (Criterion 3.4) as the second-order difference of $y_t$, which measures the local dynamics of the denoising trajectory. Implemented in a plug-and-play fashion, SADA dynamically identifies the sparsity mode {`token-wise`, `step-wise`, `multistep-wise`} at each timestep. For (b), we incorporate the gradient calculated by a specific ODE solver into both the stability criterion and the approximation correction. In particular, we derive two approximation schemes that unify the $x_0^t$ and $x_t$ trajectory, yielding a principled estimate of the per-step clean sample $x_0^t$ compatible with advanced diffusion schedulers.

Both theoretical and empirical analyses indicate that SADA is compatible with different backbone architectures (Ronneberger et al., 2015; Peebles & Xie, 2023) and solvers, and able to accelerate generative modeling in various modalities

(Chen et al., 2024c) and downstream tasks (Zhang et al., 2023) without additional training. In all tested scenarios, SADA achieves substantially better faithfulness ($\leq 0.100$ LPIPS) than existing strategies(Ye et al., 2024; Ma et al., 2024b; Liu et al., 2025a), while consistently delivering $\geq 1.8\times$ speedup. These results establish SADA as a practical plug-in for high-throughput, high-fidelity generative sampling.

In summary, the core contributions of our work are: (i) We introduce **SADA**, a training-free framework that leverages a stability criterion to adaptively sample the denoising process with principled approximation. (ii) To our best knowledge, SADA is the first paradigm that directly bridges numerical solvers of the sampling trajectory with sparsity-aware architecture optimizations. (iii) Extensive experiments conducted on various baselines, solvers, and prediction frameworks demonstrate the effectiveness of SADA to existing acceleration strategies. Moreover, SADA can be easily adapted to new generation pipelines and modalities.

## 2. Related Work

**Diffusion models** (Sohl-Dickstein et al., 2015; Ho et al., 2020; Song et al., b; Nichol & Dhariwal, 2021) have emerged as a transformative framework for generative modeling. They generate high-fidelity samples by iteratively reversing a stochastic process that gradually converts data into pure noise, delivering state-of-the-art results across a

wide range of tasks. In the realm of image synthesis, Stable Diffusion (Rombach et al., 2022) enhances computational efficiency by mapping high-dimensional pixel inputs to a compact latent space via a Variational Autoencoder (Kingma & Welling, 2014). Moreover, it incorporates text conditioning through a pre-trained text encoder (Radford et al., 2021; Oquab et al., 2024), driven by classifier-free guidance (Ho & Salimans, 2022). Building upon this foundation, later members of the latent diffusion family (Podell et al.; Chen et al., 2024b; Black-Forest-Labs, 2024) incorporate Transformers (Vaswani et al., 2017) as the primary architectural backbone, enabling improved scalability (Peebles & Xie, 2023) and higher-resolution generation (Chen et al., 2024a; Gao et al., 2024). However, these advancements come at the cost of efficiency: the *iterative nature of the denoising process* and the *quadratic complexity of self-attention* significantly slow down inference.

**Accelerating Diffusion Models**  To mitigate the above two bottlenecks, existing acceleration strategies typically focus on two fronts: *(1) reducing the number of inference steps*, and *(2) reducing the computational cost per step*.

The first paradigm involves interpreting the underlying stochastic process, developing numerical methods, and introducing novel objectives and frameworks. DDIM (Song et al., a) reformulates the diffusion process as a non-Markovian procedure. Subsequent work identifies the diffusion process as a stochastic differential equation (Song & Ermon, 2019) and converts it into solving a probabilistic-flow ordinary differential equation (PF-ODE) that could be parameterized by various model output objectives (Song et al., b; Chen et al., 2023; Kim et al.; Kingma & Gao, 2023). Building on this perspective, advanced numerical solvers significantly decrease the required number of function evaluations. Among these, the Euler Discrete Multistep (EDM) solver (Karras et al., 2022; Liu et al., 2023) has been applied effectively, while the DPM-Solver series (Lu et al., 2022a;b; Zheng et al., 2023) further enhance efficiency and stability by computing the linear component analytically with respect to the semi-linearity of PF-ODE. More recent works further explore the sampling trajectory of generative modeling. Consistency models (Song et al., 2023; Lu & Song, 2024) establish a theoretical one-step mapping from any point on the PF-ODE to the data distribution, while the flow matching (Liu et al., 2022; Albergo & Vanden-Eijnden, 2022; Lipman et al., 2023) casts generative modeling as directly learning the ODE that transports samples from the noise distribution to the data distribution.

The second paradigm mainly leverages either step-wise or token-wise sparsity in diffusion models to reduce computation overhead. On the step-wise front, DeepCache (Ma et al., 2024b) accelerates the U-Net (Ronneberger et al., 2015) based denoising model by caching high-level features

in deeper layers. Feature caching methods (Zhao et al., 2024; Ma et al., 2024a; Wimbauer et al., 2024; Zhen et al., 2025; Saghatchian et al., 2025; Liu et al., 2024; Chen et al., 2024d; Shen et al.; Liu et al., 2025b) shorten sampling latency by reusing attention outputs. PF-Diff (Wang et al., 2024) leverages previous model outputs to predict a look-ahead term in the ODE solver, analogous to the use of Nesterov momentum. These step-wise approaches typically employ a fixed schedule—determined via a hyperparameter—to guide the sparsity pattern during sampling. In parallel, token reduction strategies (Bolya & Hoffman, 2023; Kim et al., 2024; Zhen et al., 2025; Saghatchian et al., 2025) exploit the redundancy in image pixels to eliminate unnecessary tokens, thereby reducing the attention module's computational load. ToCa series (Zou et al., 2024; Zhang et al., 2024) combines caching and token pruning. DiTFastAttn (Yuan et al., 2024; Zhang et al., 2025) compresses the attention module leveraging the redundancies identified after a brief search.

Despite these advancements, none of these approaches dynamically allocate step-wise and token-wise sparsity to accelerate sampling in multi-granularity levels. Moreover, their end-to-end acceleration configurations typically hinge on specific hyperparameters or pre-trained models. Adaptive Diffusion (Ye et al., 2024) offers a promising perspective by adjusting its acceleration mode based on the prompt. However, it still requires hyperparameter tuning and does not correct for approximation error. In contrast, we frame diffusion acceleration as a stability-prediction problem rather than merely controlling error accumulation. SADA requires minimal hyperparameter tuning and incorporates principled approximation schemes that directly match our stability criterion.

## 3. Proposed Method

### 3.1. Preliminary

**Diffusion Models**  (Sohl-Dickstein et al., 2015; Ho et al., 2020; Song et al., a; Nichol & Dhariwal, 2021) formulate sample generation as an iterative denoising process. Starting from a Gaussian sample $x_T \sim \mathcal{N}(0, \mathbf{I})$, these models progressively recover a clean signal over $T$ timesteps. The forward process is defined by a variance schedule $\{\beta_t\}$ with $\alpha_t = 1 - \beta_t$ and $\bar{\alpha}_t = \prod_{i=1}^{t} \alpha_i$. The reverse (sampling) process is then modeled as:

$$p_\theta(x_{t-1} \mid x_t) = \mathcal{N}(x_{t-1}; \mu_\theta(x_t, t), \beta_t \mathbf{I}), \qquad (1)$$

where the mean $\mu_\theta(x_t, t) = \frac{1}{\sqrt{\alpha_t}} \left( x_t - \frac{1-\alpha_t}{\sqrt{1-\bar{\alpha}_t}} \epsilon_\theta(x_t, t) \right)$ is parameterized by a noise prediction model $\epsilon_\theta(x_t, t)$. Meanwhile, the data reconstruction $x_0^t$ could be calculated as:

$$x_0^t = \frac{1}{\sqrt{\bar{\alpha}_t}} \left( x_t - \sqrt{1 - \bar{\alpha}_t} \epsilon_\theta(x_t, t) \right) \qquad (2)$$

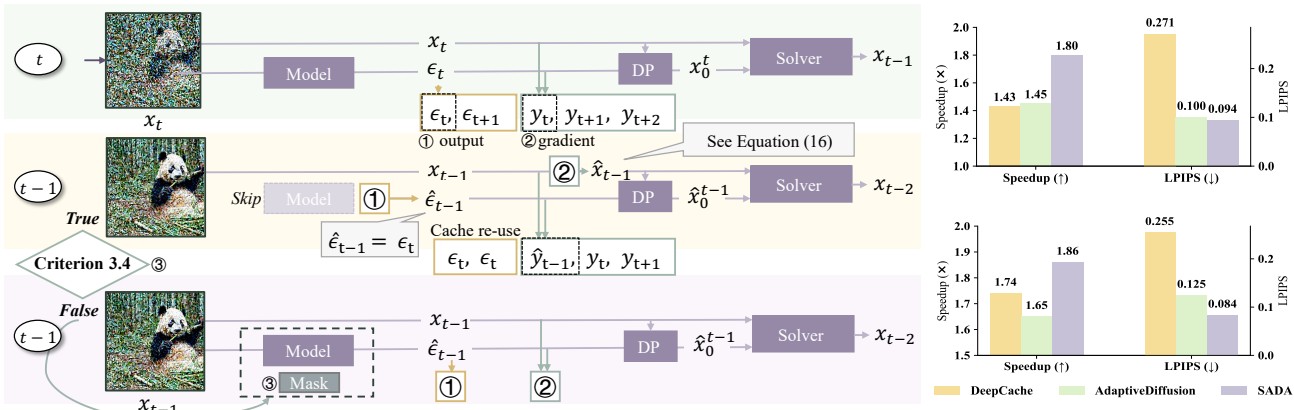

*Figure 2.* Overview paradigm of **SADA**. The sparsity mode (middle: `step-wise`, bottom: `token-wise`) at timestep $t-1$ is adaptively identified by the stability Criterion 3.4 after fresh computation at timestep $t$. Note that "DP" in the pipeline stands for "Data Prediction". **Right**: Visualization of SADA and baseline methods' performance in terms of faithfulness and efficiency. Our methods significantly outperforms existing baselines {`DeepCache`, `AdaptiveDiffusion` } on both metrics, using {`SD-2` (Top), `SDXL` (Bottom)} with DPM-solver++ 50 steps.

For text-to-image synthesis, generation typically occurs in a learned latent space (Rombach et al., 2022). In this setting, the denoising model is usually implemented using a deep neural network composed of $L$ transformer-based layers. At the $l$-th transformer layer at timestep $t$, the input latent representation is denoted by $\mathbf{x}_t^{(l)} \in \mathbb{R}^{B \times H \times W \times C}$, where $B$ is the batch size, and $H, W, C$ are the spatial and embedding dimensions, respectively.

**Diffusion ODEs** Sampling with diffusion models can be reformulated as solving a reverse-time ordinary differential equation (ODE). In particular, previous works (Song & Ermon, 2019; Song et al., b) defined the Probability-flow ODE that characterizes the continuous-time evolution of samples:

$$\frac{\mathrm{d}x}{\mathrm{d}t}\bigg|_{x=x_t} = f(t)x_t + \frac{g^2(t)}{2\sigma_t}\epsilon_\theta(x_t, t), \quad (3)$$

where $f(t) = \frac{\mathrm{d}}{\mathrm{d}t}\log\sqrt{\bar{\alpha}_t}$, $g^2(t) = \frac{\mathrm{d}\sigma_t}{\mathrm{d}t} - 2\frac{\mathrm{d}}{\mathrm{d}t}\left(\log\sqrt{\bar{\alpha}_t}\right)\sigma_t$, and $\sigma_t = \sqrt{1-\bar{\alpha}_t}$. Here, time is assumed to be continuous with $t \sim \mathcal{U}([0,1])$. Recent works explored rectified flows (Liu et al., 2022; Albergo & Vanden-Eijnden, 2022; Lipman et al., 2023), where the reverse transformation is achieved with a learned vector field $u_\theta(x_t, t)$ predicted by the denoising model:

$$\frac{\mathrm{d}x}{\mathrm{d}t}\bigg|_{x=x_t} = u_\theta(x_t, t). \quad (4)$$

Without loss of generality, we mark $\frac{\mathrm{d}x}{\mathrm{d}t}\big|_{x=x_t}$ as $y(x_t, t)$. For brevity, we further denote $y(x_t, t)$ as $y_t$, $\epsilon_\theta(x_t, t)$ as $\epsilon_t$, and $u_\theta(x_t, t)$ as $u_t$

**Step-wise Sparsity** accelerates the diffusion model by adaptively reducing the noise prediction steps. The recent

work (Ye et al., 2024) leverages a third-order difference of the latent feature map $x_t$ to identify temporal redundancy in the sampling process. Let $\Delta^{(1)}x_t = x_t - x_{t+1}$ as backward finite difference. At step $t$, given the noise $\epsilon_t$ and a threshold $\tau$, if:

$$\frac{(\|\Delta^{(1)}x_{t+2}\| + \|\Delta^{(1)}x_t\|)/2 - \|\Delta^{(1)}x_{t+1}\|}{\|\Delta^{(1)}x_{t+1}\|} \leq \tau, \quad (5)$$

then the denoising model is bypassed for the subsequent timestep $t - 1$, and $\epsilon_{t-1}$ is reused by $\epsilon_t$.

**Token-wise Sparsity** accelerates diffusion models with high representation granularity. The token indices of the input sequence in a transformer layer are partitioned into $\mathcal{I}_{\text{reduce}}$ and $\mathcal{I}_{\text{fix}}$, aiming to maximize $|\mathcal{I}_{\text{reduce}}|$ while maintaining acceptable image quality. An index mapping $I : \{1, \ldots, N\} \to \{1, \ldots, N'\}$ is defined, where $N' = |\mathcal{I}_{\text{fix}}|$. A standard token pruning (Bolya & Hoffman, 2023; Kim et al., 2024) process then discards $\mathcal{I}_{\text{reduce}}$ and retains only $\mathcal{I}_{\text{fix}}$:

$$\tilde{\mathbf{x}}[j] = \mathbf{x}[i'], \quad j = 1, \ldots, N', \quad (6)$$

where $I(i') = j$ and $i' \in \mathcal{I}_{\text{fix}}$. After self-attention $A(\cdot)$, the feature map is reconstructed by:

$$\hat{\mathbf{x}}[i] = A(\tilde{\mathbf{x}})[I(i)], \quad i = 1, \ldots, N. \quad (7)$$

Removing more tokens often degrades generation quality. Our analysis in Appendix C shows that token merging behaves as a low-pass filter, motivating our choice to build upon token pruning.

### 3.2. Overall Paradigm

Figure 2 illustrates the overall paradigm of SADA. We leverage the precise gradient from the sampling trajectory to

measure the denoising stability, as described in Sections 3.3. This Boolean measure serves as an overall criterion for our acceleration process. Specifically, when Criterion 3.4 returns `True`, we apply step-wise cache-assisted pruning (refer to Section 3.4) with dual approximation scheme {`step-wise`, `multistep-wise`}. Conversely, when Criterion 3.4 returns `False`, we apply token-wise cache-assisted pruning (refer to Section 3.5).

### 3.3. Modeling the Stability of the Denoising Process

We formulate the acceleration of the denoising process as a stability prediction problem. Given a pre-trained diffusion architecture with noise prediction $\epsilon_t$ at timestep $t$, we seek a dynamic criterion for step-wise and token-wise cache-assisted pruning (i.e., optimal sparsity allocation) that reduces computation at timestep $t-1$ without degrading sample fidelity.

The criterion for acceleration should match the specific approximation method that mitigates the loss resulting from step-wise pruning. AdaptiveDiffusion (Ye et al., 2024) directly caches and reuses the noise across steps. However, approximating $\hat{\epsilon}_{t-1} = \epsilon_t$ introduces a mismatch between the sample state $x_{t-1}$ and its corresponding noise, causing error to accumulate. A more principled approach should incorporate *historical information* to correct this approximation.

Concretely, we propose a *third-order extrapolation* of the sample state at $t-1$, which implicitly carries historical noise along the ODE trajectory. A straightforward third-order backward finite-difference baseline can be written as $\hat{x}_{t-1} = 3x_t - 3x_{t+1} + x_{t+2}$. This yields a correction term, $x_{t-1} - \hat{x}_{t-1} = \Delta^{(3)}x_{t-1}$. Moreover, $\hat{x}_{t-1}$ is theoretically bounded by Theorem 3.1, which guarantees its stability under small $\Delta t$.

**Theorem 3.1.** *Let $f \in C^k[a,b]$ be a smooth function and let $x_0 := x$, with equally spaced grid points $x_i := x + ih$ for $i = 0, 1, \ldots, k-1$. Define $P_{k-1}(t)$ as the degree-$(k-1)$ Lagrange interpolant of $f$ at $\{x_i\}_{i=0}^{k-1}$.*

*Then, the extrapolated value at $x - h$ satisfies:*

$$f(x-h) = \sum_{i=0}^{k-1} \alpha_i f(x_i) + R_k(h), \qquad (8)$$

*where the weights are given by:*

$$\alpha_i = (-1)^i \binom{k}{i+1}, \quad i = 0, 1, \ldots, k-1. \quad (9)$$

*The error bound of the remainder term is:*

$$R_k(h) = \mathcal{O}(h^k). \qquad (10)$$

To derive our stability criterion, we now present the following supplemental theorems:

**Theorem 3.2.** *The expected value of $x_t$ over the joint distribution of $x_0$, $\epsilon$, and timestep $t$ satisfies: $\mathbb{E}_{x_0,\epsilon,t}[x_t] = \sqrt{\bar{\alpha}_t} \cdot \mathbb{E}_{x_0}[x_0]$.*

(i) From Theorem 3.2, we know that the trajectory $\{x_t\}$ is continuous in expectation. Consequently, the empirical average $x_t$ also exhibits continuity by the law of large numbers (Feller et al., 1971).

**Theorem 3.3.** *Let the network $\epsilon_\theta(x, t)$ be trained with the standard mean-squared error (MSE) objective:*

$$\mathcal{L}(\theta) = \mathbb{E}_{x_0,\epsilon,t}\left[\|\epsilon - \epsilon_\theta(x_t, t)\|^2\right], \qquad (11)$$

*Suppose $\theta^\star$ minimizes $\mathcal{L}$, and the training is sufficiently converged. Then, following **Assumption 1** that $\epsilon_\theta(x_t, t)$ is Lipschitz in $x_t$ and $t$, we have the following consistency property for sampling-time inputs $\hat{x}_t$:*

$$\mathbb{E}_{x_0,\epsilon,t}[\epsilon - \epsilon_{\theta^\star}(\hat{x}_t, t)] \to 0 \quad as \ \|\hat{x}_t - x_t\| \to 0. \quad (12)$$

(ii) From Theorem 3.3, a well-trained denoiser permits us to treat the sampling trajectory as a continuous process, preserving structural consistency.

We therefore assume that, in a stable regime, the sign of consecutive third-order differences *remains consistent*: $\text{sign}(\Delta^{(3)}x_t) = \text{sign}(\Delta^{(3)}x_{t-1})$. Consequently, if the extrapolation error $x_{t-1} - \hat{x}_{t-1}$ is aligned with the true curvature $\Delta^{(3)}x_{t-1}$, then $\hat{x}_{t-1}$ serves as a directionally accurate approximation. To incorporate precise gradient information, we leverage the always-hold identity $\Delta^{(2)}y_t \cdot \Delta^{(3)}x_t < 0$ by construction. Combining the sign measure with the identity, we substitute $\Delta^{(3)}x_t$ with $\Delta^{(3)}x_{t-1} = x_{t-1} - \hat{x}_{t-1}$ and yield the following criterion:

**Criterion 3.4.** *A timestep $t$ is considered stable and eligible for acceleration if the extrapolation error is anti-aligned with the local curvature of velocity:*

$$(x_{t-1} - \hat{x}_{t-1}) \cdot \Delta^{(2)}y_t < 0. \qquad (13)$$

This stability measure ensures that the extrapolated state $\hat{x}_{t-1}$ lies in the correct direction with respect to curvature correction.

### 3.4. Step-wise Cache-Assisted Pruning

This section proposes two complementary approximation schemes for Step-wise Cache-Assisted Pruning: (`step-wise`) and (`multistep-wise`). Either approximation scheme produces a clean-sample estimate $\hat{x}_0^t$, which is then fed into advanced samplers (e.g., DPM-Solver++ or

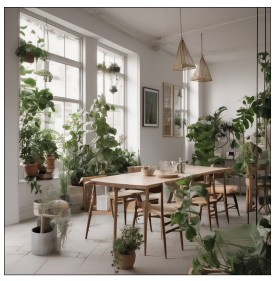 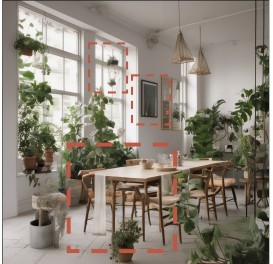 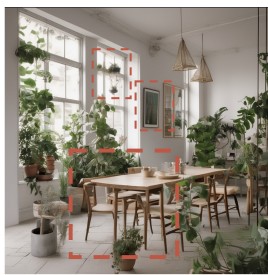 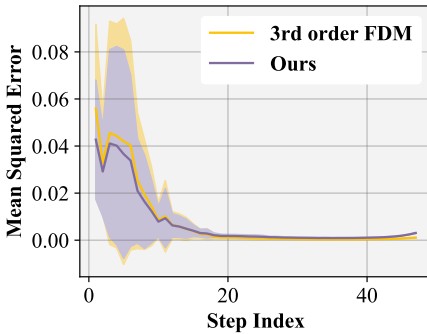

**Baseline** (SDXL)    **3rd Order FDM** (LPIPS = 0.072)    **3rd Order AM** (LPIPS = 0.054)

*Figure 3.* Comparison of $x_t$-approximation strategies. Left: Step-wise pruning result of the third-order finite difference method and third-order Adams-Moulton method. Right: Mean Squared Error comparison between two strategies, averaged over 50 randomly selected prompts. Shaded regions indicate the standard deviation at each step.

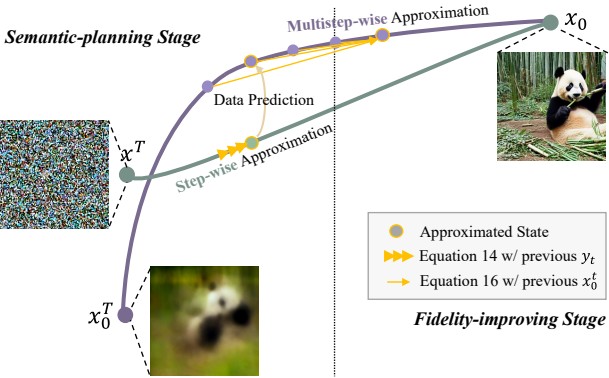

*Figure 4.* Illustration of the proposed dual approximation scheme on the $x_0^t$ and $x_t$ trajectory.

EDM (Karras et al., 2022; Lu et al., 2022a)). This unified framework aligns the $x_0^t$ and $x_t$ trajectories (Fig. 4). Below, we detail the design choices for each approximation.

**Step-wise Approximation** As noted in Section 3.3, a simple baseline for approximating $x_{t-1}$ is a third-order backward finite difference. Empirical experiments demonstrate low reconstruction errors, as shown in Figure 3. Note that we reuse the noise prediction at step $t$, formulated as $\hat{\epsilon}_{t-1} \leftarrow \epsilon_t$, under this scheme. However, since we have exact derivative information $y_t$ (See Eq. 3 and 4) we adopt a third-order **Adams–Moulton** method (Iserles, 2009) along the ODE trajectory:

**Theorem 3.5.** *Using the second- and third-order Adams–Moulton method, we define the estimator:*

$$\hat{x}_{t-1} := x_t - \frac{5\Delta t}{6}y_t - \frac{5\Delta t}{6}y_{t+1} + \frac{2\Delta t}{3}y_{t+2}, \quad (14)$$

*whose local truncation error satisfies:* $\hat{x}_{t-1} - x_{t-1} = \mathcal{O}(\Delta t^2)$.

The full derivation and the corresponding error bound are provided in Proposition B.1, Theorems 3.1, B.2, and 3.5 in the Appendix B.2. To quantify the benefit of these precise updates, we measure per-step reconstruction error on 50 randomly sampled MS-COCO prompts (Lin et al., 2014). Our Adams–Moulton scheme results in lower mean error and smaller standard deviation, compared to third-order finite difference, as illustrated in Figure 3.

Given that $x_0^t$ captures structural information and serves as the initial input to all schedulers, accurately reconstructing $x_0^t$ is critical. Theorem 3.6 establishes an upper bound on the reconstruction error at timestep $t$, based on the third-order estimator in Theorem 3.5 and incorporating the effect of noise reuse.

**Theorem 3.6.** *Let $\hat{x}_0^t$ denote the reconstruction of $x_0$ at time $t$. Then, the final reconstruction $\hat{x}_0^t$ satisfies the following error bound:*

$$\|\hat{x}_0^t - x_0^t\| = \mathcal{O}(\Delta t) + \mathcal{O}(\Delta x_t). \quad (15)$$

This theorem characterizes the reconstruction error as a first-order term in both the scheduler resolution $\Delta t$ and the variation $\Delta x_t$. The detailed proof can be found in Appendix B.2.

**Multistep-wise Approximation** Prior work (Liu et al., 2025b) empirically divides the denoising trajectory into a *semantic-planning* stage and a subsequent *fidelity-improving* stage, corresponding to regions where data $x_0^t$ is inherently stable (see Figure 4). In these stable regions, one can safely use larger effective step sizes by compensating with higher-order interpolation. Building on this insight, we implement a uniform step-wise pruning strategy with Lagrange interpolation, once the trajectory enters the stable regime.

For example, consider a 50-step process. To achieve a step-wise pruning interval of 4 after stabilization (i.e., compute every 4th step fully and interpolate the skipped steps via

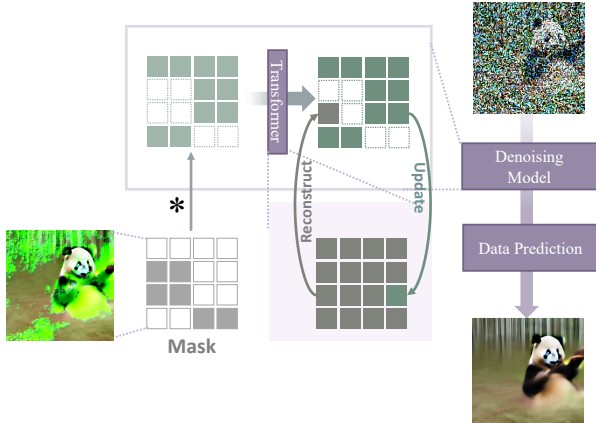

*Figure 5.* Illustration of the proposed Token-wise strategy. The criterion at token-level generates a "mask" to guide the pruning process at the input of the $l$-th attention layer. The pruned feature map is then reconstructed at the output of the layer using its cached representation $\mathcal{C}_l$.

Lagrange), we store $x_0^t$ every 4 steps before stabilization. Their indices define the fixed-size set $I$, which is a rolling buffer to limit memory usage. For any skipped $t$:

**Theorem 3.7.** *Let $I = \{0, 1, \ldots, k\}$ be $k + 1$ distinct indexes with known cached values $\{x_0^{t_i}\}_{i \in I}$. For any skipped timestep $t \notin \{t_i\}_{i \in I}$, define the interpolated reconstruction as:*

$$\hat{x}_0^t := \sum_{i \in I} \left( \prod_{j \in I \setminus \{i\}} \frac{t - t_j}{t_i - t_j} \right) x_0^{t_i}. \tag{16}$$

*Then, under the assumption that $x_0^\tau$ is $(k + 1)$-times continuously differentiable over $\tau \in [t_{\min}, t_{\max}]$, the interpolation error satisfies:*

$$\|\hat{x}_0^t - x_0^t\| = \mathcal{O}(h^{k+1}), \tag{17}$$

*where $h$ is the maximum step spacing among $\{t_i\}$.*

This multi-step cache-assisted pruning strategy significantly induce step-wise sparsity while preserving sample fidelity.

### 3.5. Token-wise Cache-Assisted Pruning

At timestep $t$, if the Criterion 3.4 returns `False`, we continue to evaluate our stability measure under a high granularity level. The core idea of our token-wise algorithm is aligned with its step-wise counterpart: (i) **fix** unstable tokens for full calculation (ii) **reduce** the stable token and approximate it by previous representation in latent cache. This procedure partitions the tokens into two sets, $\mathcal{I}_{\text{fix}}$ (unstable tokens) and $\mathcal{I}_{\text{reduce}}$ (stable tokens), which define our adaptive pruning configuration for the subsequent timestep.

Our algorithm is formulated as follows:

(i) *Cache Initialization*: Let $T^*$ denote the starting timestep for Cache-Assisted Pruning and $i$ the caching interval. For an input at timestep $t - 1$ and transformer layer $l$, if $(t - 1 - T^*) \mod i = 0$, we initialize the cache after a full computation:

$$\mathcal{C}_l = A(\mathbf{x}_{t-1}^{(l)}) \in \mathbb{R}^{B \times N \times C} \tag{18}$$

where $\mathcal{C}_l$ is the feature map stored in the cache for layer $l$, and it is updated throughout the denoising process.

(ii) *Cache Update*: When $(t - 1 - T^*) \mod i \neq 0$, we prune the input data into $\tilde{\mathbf{x}}_{t-1}^{(l)} \in \mathbb{R}^{B \times N' \times C}$ where its length $N' = |\mathcal{I}_{\text{fix}}|$. After the Transformer (/Attention), we update $\mathcal{C}_l$ with fresh tokens:

$$\mathcal{C}_l[i] = A(\tilde{\mathbf{x}}_{t-1}^{(l)})[I(i)] \quad \text{for } i \in \mathcal{I}_{\text{fix}}. \tag{19}$$

(iii) *Cache-Assisted Reconstruction*: The pruned tokens are approximated by their cached representations.:

$$\hat{\mathbf{x}}_{t-1}^{(l)}[i] = \begin{cases} A(\tilde{\mathbf{x}}_{t-1}^{(l)})[I(i)], & \text{if } i \in \mathcal{I}_{\text{fix}}, \\ \mathcal{C}_l[i], & \text{if } i \in \mathcal{I}_{\text{reduce}}. \end{cases} \tag{20}$$

We keep the reconstructed sequence $\hat{\mathbf{x}}_{t-1}^{(l)}$ synchronized with $\mathcal{C}_l$ for subsequent timesteps.

## 4. Experiments

### 4.1. Experiment Settings

**Model Configurations** To verify the generalization of the proposed approach, we evaluate a set of widely used text-to-image models employing different backbone architectures: SD-2 (U-Net), SDXL (Podell et al.) (modified U-Net), and Flux.1-dev (Black-Forest-Labs, 2024) (DiT). We perform evaluations using two sampling schedulers—Euler Discrete Multistep (EDM) (Karras et al., 2022) Solver (first-order) and DPM-Solver++ (Lu et al., 2022b) (second-order) —each configured with 50 sampling steps. All pipelines are implemented using the Huggingface Diffusers framework. Experiments with Flux.1-dev are executed on a single NVIDIA A100 GPU, while the remaining experiments are run on a single NVIDIA A5000 GPU.

**Evaluation Metrics** We compare our proposed paradigm with widely-adopted training-free acceleration strategies, DeepCache, AdaptiveDiffusion, and TeaCache. (Ma et al., 2024b; Ye et al., 2024; Liu et al., 2025a). DeepCache caches and reuses the latent feature in the middle layer of the U-Net architecture. AdaptiveDiffusion skips the noise predictor and reuses the previous predicted noise guided by a third-order estimator. TeaCache introduces a caching threshold

*Table 1.* Quantitative results on MS-COCO 2017 (Lin et al., 2014).

| Model | Scheduler | Methods | PSNR ↑ | LPIPS ↓ | FID ↓ | Speedup Ratio |
|-------|-----------|---------|--------|---------|-------|---------------|
| SD-2 | DPM++ | DeepCache | 17.70 | 0.271 | 7.83 | 1.43× |
| | | AdaptiveDiffusion | 24.30 | 0.100 | 4.35 | 1.45× |
| | | SADA | **26.34** | **0.094** | **4.02** | **1.80×** |
| | Euler | DeepCache | 18.90 | 0.239 | 7.40 | 1.45× |
| | | AdaptiveDiffusion | 21.90 | 0.173 | 7.58 | **1.89×** |
| | | SADA | **26.25** | **0.100** | **4.26** | 1.81× |
| SDXL | DPM++ | DeepCache | 21.30 | 0.255 | 8.48 | 1.74× |
| | | AdaptiveDiffusion | 26.10 | 0.125 | 4.59 | 1.65× |
| | | SADA | **29.36** | **0.084** | **3.51** | **1.86×** |
| | Euler | DeepCache | 22.00 | 0.223 | 7.36 | **2.16×** |
| | | AdaptiveDiffusion | 24.33 | 0.168 | 6.11 | 2.01× |
| | | SADA | **28.97** | **0.093** | **3.76** | 1.85× |
| Flux | Flow-matching | TeaCache | 19.14 | 0.216 | 4.89 | 2.00× |
| | | SADA | **29.44** | **0.060** | **1.95** | **2.02×** |

*Table 2.* Ablation study on few-step sampling across schedulers. Results on MS-COCO 2017.

| **SD-2** | | | | | |
|----------|-------|--------|---------|-------|---------|
| Scheduler | Steps | PSNR ↑ | LPIPS ↓ | FID ↓ | Speedup |
| DPM++ | 50 | 26.34 | 0.094 | 4.02 | 1.80× |
| | 25 | 28.15 | 0.073 | 3.13 | 1.48× |
| | 15 | 29.84 | 0.072 | 3.05 | 1.24× |
| Euler | 50 | 26.25 | 0.100 | 4.26 | 1.81× |
| | 25 | 26.83 | 0.088 | 3.87 | 1.48× |
| | 15 | 29.34 | 0.076 | 3.70 | 1.25× |
| **SDXL** | | | | | |
| Scheduler | Steps | PSNR ↑ | LPIPS ↓ | FID ↓ | Speedup |
| DPM++ | 50 | 29.36 | 0.084 | 3.51 | 1.86× |
| | 25 | 30.84 | 0.073 | 2.80 | 1.52× |
| | 15 | 31.91 | 0.073 | 2.54 | 1.29× |
| Euler | 50 | 28.97 | 0.093 | 3.76 | 1.85× |
| | 25 | 29.42 | 0.085 | 3.13 | 1.50× |
| | 15 | 31.28 | 0.084 | 3.26 | 1.26× |

that measures the error accumulation. All experiments are conducted using the MSCOCO-2017 validation set as generation prompts under identical conditions to assess efficiency and quality. We report speedup ratios compared to the baseline as a measure of generation efficiency. Generation quality is evaluated using the Peak Signal-to-Noise Ratio (PSNR), Learned Perceptual Image Patch Similarity (LPIPS), and Fréchet Inception Distance (FID) between original generated and accelerated samples.

### 4.2. Main Results

As shown in Table 1, SADA consistently outperforms Deep-Cache, AdaptiveDiffusion, and TeaCache across all settings. Compared to other acceleration strategies, it drives

FID down from 8.48 to 3.51 (59% ↓) on SDXL with DPM-Solver++, and on Flux.1-dev from 4.89 to 1.95 (60% ↓). Across every configuration, SADA maintains LPIPS ≤ 0.100 relative to the original samples—substantially better than competing methods—demonstrating only negligible perceptual deviation from the unmodified baseline. Crucially, these significant quality improvements incur no extra compute beyond the acceleration itself, yielding consistent 1.8–2× speedups, outperforming the competing methods in the majority of cases, underscoring SADA's effectiveness as a loss-free acceleration framework.

### 4.3. Ablation Studies

We provide justification for our choice of base step $T = 50$ in Figure A.3. To evaluate SADA's performance under few-step sampling, Table 2 reports results on {SD-2, SDXL} using {Euler, DPM-Solver++} while varying the number of inference steps {50, 25, 15}. As the step count decreases, SADA achieves higher similarity: PSNR rises from 26.25 dB to 29.34 dB, FID falls from 4.26 to 3.70, and LPIPS drops from 0.100 to 0.076. We attribute this trend to reduced error accumulation when using fewer steps. Meanwhile, SADA can further accelerate sampling under a few-step setting. The speedup ratio maintains $\sim 1.5\times$ under 25 steps denoising, and $\sim 1.25\times$ under 15 steps denoising, highlighting SADA's effectiveness under a low computational budget. Note that the Lagrange interpolation parameters are slightly adjusted to match the shorter denoising schedules in these few-step settings.

### 4.4. Downstream Tasks and Data Modalities

This section demonstrates that SADA has potential to accelerate any generative modeling with an iterative generative process, regardless of the downstream task or data modality.

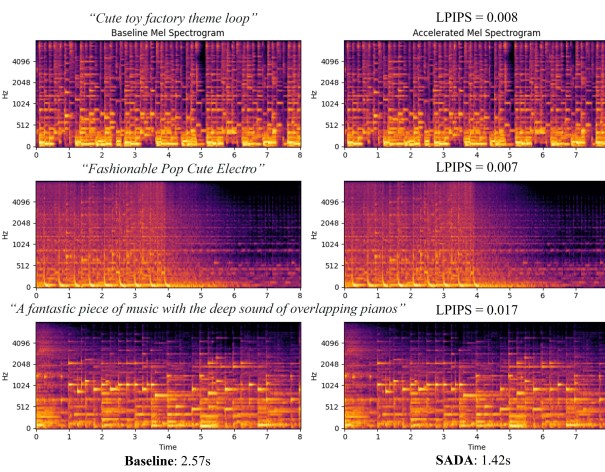

*Figure 6.* SADA deployment on MusicLDM on different text prompts. SADA accelerates MusicLDM by $\sim 1.81\times$ while maintaining the spectrogram LPIPS under 0.020.

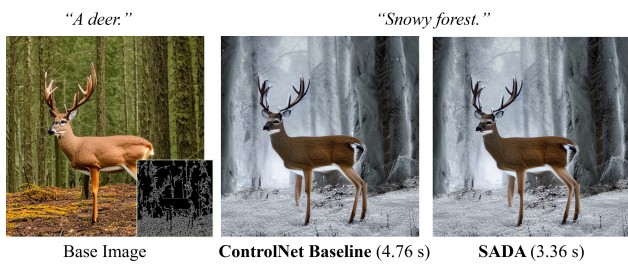

*Figure 7.* SADA deployment on ControlNet. We demonstrate the SD-1.5-based ControlNet pipeline trained on canny edges as conditional input. SADA accelerates ControlNet by $\sim 1.41\times$ while preserving fidelity.

**Data Modality**  We evaluate SADA on music and audio generation using the MusicLDM (Chen et al., 2024c) pipeline to synthesize 8-second clips. We compare both perceptual quality and spectrogram similarity between accelerated samples and the unmodified baseline. As shown in Figure 6, SADA achieves an LPIPS of $\sim 0.010$ between accelerated and baseline audio, while reducing sampling time by $\sim 1.81\times$. This implies that SADA has the potential to implemented in future baselines models for any-modality generation with little to no modifications.

**Downstream Task**  To validate cross-pipeline compatibility, we apply SADA directly on top of ControlNet (Zhang et al., 2023) without any additional fine-tuning or architectural changes. Figure 7 demonstrates that SADA preserves faithfulness between ControlNet-conditioned outputs while accelerating the sampling by $\sim 1.41\times$. This implies that SADA could be seamlessly deployed in professional workflows.

## 5. Conclusion

We present Stability-guided Adaptive Diffusion Acceleration (SADA), a training-free paradigm that adaptively accelerates the sampling process of ODE-based generative models (mainly Diffusion and Flow-matching). Leveraging trajectory gradient calculated from the numerical solver, SADA dynamically exploits step-wise and token-wise sparsity during the procedure with principled and error-bounded approximation, bridging between the sampling trajectories and sparsity-aware optimizations. Extensive experiments on SD-2, SDXL, and Flux across EDM and DPM++ solvers both demonstrate consistent $\geq 1.8\times$ speedups with negligible fidelity loss (LPIPS $\leq 0.10$, FID $\leq 4.5$) compared to unmodified baselines, significantly outperforming existing strategies. Moreover, we show that SADA generalizes across modalities—achieving $\sim 1.81\times$ acceleration on MusicLDM and $\sim 1.41\times$ on ControlNet—without additional tuning.

## Acknowledgment

This material is based upon work supported by the U.S. National Science Foundation under award No.2112562. This work is also supported by ARO W911NF-23-2-0224 and NAIRR Pilot project NAIRR240270. Any opinions, findings and conclusions or recommendations expressed in this material are those of the author(s) and do not necessarily reflect the views of the U.S. National Science Foundation, ARO, NAIRR, and their contractors. In addition, we thank the area chair and reviewers for their valuable comments.

## Impact Statement

This paper aims to advance the field of adaptive acceleration of generative models. We believe that our work has significant potential impact for the deployment and inference of generative models in any modalities. While there are many potential societal consequences of our work, none which we feel must be specifically highlighted here.

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

# A. Appendix

Due to the page limit of the main manuscript, we provide more theoretical and implementation details in the appendix, organized as follows:

- Sec. B: **Mathematical Foundations**
  - Sec. B.1: Theoretical Assumptions.
  - Sec. B.2: Proofs of Main Theorems.

- Sec. C: **Analysis on Existing Token Reduction Methods**.

- Sec. D: **Additional Experimentss**.

# B. Mathematical Foundations

## B.1. Theoretical Assumptions

**1** The network output $\epsilon_\theta(x_t, t)$ is jointly Lipschitz continuous in both $x_t$ and $t$. In our experiments, we skip the first and last time steps to avoid potential issues with infinite Lipschitz (Yang et al., 2023) constants near the boundaries.

## B.2. Proofs of Main Theorems

**Theorem 3.2.** Let $x_t = \sqrt{\bar{\alpha}_t}x_0 + \sqrt{1 - \bar{\alpha}_t}\,\epsilon$, where $\epsilon \sim \mathcal{N}(0, I)$ and $x_0 \sim p(x_0)$ is independent of $\epsilon$. Then the expected value of $x_t$ over the joint distribution of $x_0$, $\epsilon$, and timestep $t$ satisfies

$$\mathbb{E}_{x_0, \epsilon, t}[x_t] = \sqrt{\bar{\alpha}_t} \cdot \mathbb{E}_{x_0}[x_0]. \tag{A.1}$$

*Proof.* By the definition of the forward process in diffusion models, the latent variable at timestep $t$ is given by:

$$x_t = \sqrt{\bar{\alpha}_t}x_0 + \sqrt{1 - \bar{\alpha}_t}\,\epsilon. \tag{A.2}$$

Taking expectation over $x_0 \sim p(x_0)$, $\epsilon \sim \mathcal{N}(0, I)$, and $t \sim \text{Uniform}(\{1, \ldots, T\})$, we get:

$$\mathbb{E}_{x_0, \epsilon, t}[x_t] = \mathbb{E}_t \left[ \mathbb{E}_{x_0}[\sqrt{\bar{\alpha}_t}x_0] + \mathbb{E}_\epsilon[\sqrt{1 - \bar{\alpha}_t}\,\epsilon] \right]. \tag{A.3}$$

Since $\epsilon \sim \mathcal{N}(0, I)$, we have $\mathbb{E}[\epsilon] = 0$, and thus:

$$\mathbb{E}_{x_0, \epsilon, t}[x_t] = \mathbb{E}_t \left[ \sqrt{\bar{\alpha}_t} \cdot \mathbb{E}_{x_0}[x_0] \right] = \sqrt{\bar{\alpha}_t} \cdot \mathbb{E}_{x_0}[x_0]. \tag{A.4}$$

This concludes the proof. □

**Theorem 3.3** (Consistency of the network estimator under trajectory approximation)**.** Let the network $\epsilon_\theta(x, t)$ be trained with the standard mean-squared error (MSE) objective:

$$\mathcal{L}(\theta) = \mathbb{E}_{x_0, \epsilon, t} \left[ \|\epsilon - \epsilon_\theta(x_t, t)\|^2 \right], \quad x_t = \sqrt{\bar{\alpha}_t}x_0 + \sigma_t\epsilon. \tag{A.5}$$

Suppose $\theta^\star$ minimizes $\mathcal{L}$, and the training is sufficiently converged. Then, following **Assumption 1** that $\epsilon_\theta(x_t, t)$ is

Lipschitz in $x_t$ and $t$, we have the following consistency property for sampling-time inputs $\hat{x}_t$:

$$\mathbb{E}_{x_0,\epsilon,t}[\epsilon - \epsilon_{\theta^\star}(\hat{x}_t, t)] \to 0 \quad \text{as } \|\hat{x}_t - x_t\| \to 0. \tag{A.6}$$

*Proof.* Since $x_t$ is a linear deterministic combination of $(x_0, \epsilon)$ at time step $t$, the distribution $p(x_t, t)$ is induced from the joint distribution of $x_0, \epsilon$, and $t$. Therefore, minimizing the MSE loss over $(x_0, \epsilon, t)$ is equivalent to minimizing it over $(x_t, t)$. The optimal solution in the $L^2$ sense is the conditional expectation:

$$\epsilon_{\theta^\star}(x_t, t) = \mathbb{E}_{x_0,\epsilon,t|x_t,t}[\epsilon \mid x_t, t] \tag{A.7}$$

Taking expectation again:

$$\mathbb{E}_{x_0,\epsilon,t}[\epsilon - \epsilon_{\theta^\star}(x_t, t)] = 0. \tag{A.8}$$

Now, for any approximation $\hat{x}_t$, by Lipschitz continuity:

$$\|\epsilon_{\theta^\star}(\hat{x}_t, t) - \epsilon_{\theta^\star}(x_t, t)\| \le L\|\hat{x}_t - x_t\|. \tag{A.9}$$

Thus,

$$\|\mathbb{E}_{x_0,\epsilon,t}[\epsilon - \epsilon_{\theta^\star}(\hat{x}_t, t)]\| \le \|\mathbb{E}_{x_0,\epsilon,t}[\epsilon - \epsilon_{\theta^\star}(x_t, t)]\| + \mathbb{E}_{x_0,\epsilon,t}\left[\|\epsilon_{\theta^\star}(x_t, t) - \epsilon_{\theta^\star}(\hat{x}_t, t)\|\right]. \tag{A.10}$$

The first term is zero; the second vanishes as $\hat{x}_t \to x_t$.

Therefore,

$$\mathbb{E}_{x_0,\epsilon,t}[\epsilon - \epsilon_{\theta^\star}(\hat{x}_t, t)] \to 0, \tag{A.11}$$

as $\|\hat{x}_t - x_t\| \to 0$. $\square$

**Theorem 3.7** (Lagrange Interpolation for Cache-Assisted Reconstruction). Let $I = \{0, 1, \dots, k\}$ be $k + 1$ distinct indexes with known cached values $\{x_0^{t_i}\}_{i \in I}$. For any skipped timestep $t \notin \{t_i\}_{i \in I}$, define the interpolated reconstruction as:

$$\hat{x}_0^t := \sum_{i \in I}\left(\prod_{j \in I \setminus \{i\}} \frac{t - t_j}{t_i - t_j}\right) x_0^{t_i}. \tag{A.12}$$

Then, under the assumption that $x_0^\tau$ is $(k + 1)$-times continuously differentiable over $\tau \in [t_{\min}, t_{\max}]$, the interpolation error satisfies:

$$\|\hat{x}_0^t - x_0^t\| = \mathcal{O}(h^{k+1}), \tag{A.13}$$

where $h$ is the maximum step spacing among $\{t_i\}$.

*Proof.* The expression in Equation A.12 is the Lagrange interpolation formula for approximating a function at an unobserved point $t$ using $k + 1$ known values $\{x_0^{t_i}\}$ at points $\{t_i\} \in I$.

By classical interpolation theory, the interpolation error at $t$ for a $(k + 1)$-times continuously differentiable function $x_0^\tau$ satisfies:

$$x_0^t - \hat{x}_0^t = \frac{1}{(k+1)!}\frac{d^{k+1}x_0^\xi}{d\xi^{k+1}} \cdot \prod_{i \in I}(t - t_i), \quad \text{for some } \xi \in [t_{\min}, t_{\max}]. \tag{A.14}$$

Taking norm and bounding the derivative gives:

$$\|\hat{x}_0^t - x_0^t\| \leq \frac{1}{(k+1)!} \max_\xi \left\| \frac{d^{k+1} x_0^\xi}{d\xi^{k+1}} \right\| \cdot \prod_{i \in I} |t - t_i|. \tag{A.15}$$

If all time steps are approximately uniformly spaced with step size $h$, then $|t - t_i| \leq Ch$ for some constant $C$, and hence:

$$\|\hat{x}_0^t - x_0^t\| = \mathcal{O}(h^{k+1}), \tag{A.16}$$

as claimed. $\square$

**Theorem 3.1** (Backward Extrapolation via Lagrange Interpolation). Let $f \in C^k[a, b]$ be a smooth function and let $x_0 := x$, with equally spaced grid points $x_i := x + ih$ for $i = 0, 1, \ldots, k - 1$. Define $P_{k-1}(t)$ as the degree-$(k-1)$ Lagrange interpolant of $f$ at $\{x_i\}_{i=0}^{k-1}$.

Then, the extrapolated value at $x - h$ satisfies:

$$f(x - h) = \sum_{i=0}^{k-1} \alpha_i f(x_i) + R_k(h), \tag{A.17}$$

where the weights are given by:

$$\alpha_i = (-1)^i \binom{k}{i+1}, \quad i = 0, 1, \ldots, k - 1, \tag{A.18}$$

and the remainder term is:

$$R_k(h) = \frac{f^{(k)}(\xi)}{k!} \prod_{j=0}^{k-1} (x - h - x_j), \qquad \xi \in [x - h, x + (k-1)h]. \tag{A.19}$$

and the error bound of the remainder term is:

$$R_k(h) = \mathcal{O}(h^k). \tag{A.20}$$

*Proof.* Let $x_i := x + ih$ for $i = 0, \ldots, k - 1$. We construct the Lagrange interpolating polynomial:

$$P_{k-1}(t) = \sum_{i=0}^{k-1} f(x_i)\, \ell_i(t), \tag{A.21}$$

where the Lagrange basis functions are:

$$\ell_i(t) = \prod_{\substack{j=0 \\ j \neq i}}^{k-1} \frac{t - x_j}{x_i - x_j}. \tag{A.22}$$

Evaluating at $t = x - h$ gives:

$$f(x - h) = P_{k-1}(x - h) + R_k(h), \tag{A.23}$$

where the remainder term is the standard Lagrange error:

$$R_k(h) = \frac{f^{(k)}(\xi)}{k!} \prod_{j=0}^{k-1} (x - h - x_j), \quad \xi \in [x - h, x + (k-1)h]. \tag{A.24}$$

Since $x_j = x + jh$, we have: $R_k(h) = \mathcal{O}(h^k)$. Now compute the weights $\alpha_i := \ell_i(x - h)$, we have:

$$\ell_i(x - h) = \prod_{\substack{j=0 \\ j \neq i}}^{k-1} \frac{(x - h) - (x + jh)}{(x + ih) - (x + jh)} = \prod_{\substack{j=0 \\ j \neq i}}^{k-1} \frac{-(j+1)h}{(i-j)h}. \tag{A.25}$$

Canceling $h$ and simplifying signs:

$$\ell_i(x - h) = (-1)^{k-1} \cdot \frac{(k-1)!}{i!(k-1-i)!} \cdot \frac{1}{i+1} = (-1)^i \binom{k}{i+1}. \tag{A.26}$$

Therefore:

$$f(x - h) = \sum_{i=0}^{k-1} (-1)^i \binom{k}{i+1} f(x_i) + R_k(h), \tag{A.27}$$

as claimed. $\qquad \square$

---

**Proposition B.1** (High-order backward difference as weighted combination). *Let $\alpha_i := (-1)^i \binom{k}{i+1}$ be the extrapolation coefficients in Theorem 3.1. Then the following linear combination defines the k-th order backward finite difference:*

$$f(x - h) - \sum_{i=0}^{k-1} \alpha_i f(x + ih) = \Delta^{(k)} f(x - h), \tag{A.28}$$

*where $\Delta^{(k)}$ is the standard k-th order backward difference operator:*

$$\Delta^{(k)} f(x - h) := \sum_{i=0}^{k} (-1)^i \binom{k}{i} f(x + ih). \tag{A.29}$$

---

*Proof.* The result follows directly by substituting the expression for $\alpha_i$ from Theorem 3.1 into Equation (A.28) and matching terms with the standard definition of $\Delta^{(k)}$. $\qquad \square$

---

**Theorem B.2** (Adams-Moulton Method via Forward Lagrange Quadrature). *Let $y(x) \in C^k[a, b]$ and let $x_0 := x$ with equally spaced grid points $x_i := x + ih$ for $i = 0, 1, \ldots, k - 1$. Let $P_{k-1}(t)$ be the degree-$(k-1)$ Lagrange interpolant of $f$, which is derivative of $y$, at $\{x_i\}_{i=0}^{k-1}$.*

*Define the one-step approximation of the integral:*

$$y(x - h) = y(x) - \int_{x-h}^{x} f(s) ds. \tag{A.30}$$

*Then, the **Adams-Moulton method of order** $k$ is:*

$$\hat{y}_{n-1} = y_n - h \left( \sum_{i=0}^{k-1} \beta_i f_{n+i} + \beta_{-1} f_{n-1} \right), \tag{A.31}$$

*where: $y_{n+j} = y(x_j)$, $f_{n+j} = f(x_j)$, and the quadrature weights $\beta_{-1}, \beta_0, \ldots, \beta_{k-1}$ are given by:*

---

$$\beta_j := \int_0^1 \ell_j(s)\, ds, \quad \textit{for } j = -1, 0, \ldots, k-1, \tag{A.32}$$

*where $\ell_j(s)$ are the Lagrange basis polynomials defined on nodes $\tau_j = j$, with $\tau_{-1} = -1$ for the future value $y_{n-1}$.*

*The local truncation error is:*

$$\hat{y}_{n-1} - y_{n-1} = \mathcal{O}(h^{k+1}). \tag{A.33}$$

*Proof.* We aim to approximate the integral:

$$y(x - h) = y(x) - \int_{x-h}^x f(s)\, ds. \tag{A.34}$$

Let us define a variable substitution $s = x - h + \tau h$, so that:

$$\int_{x-h}^x f(s)\, ds = h \int_0^1 f(x - h + \tau h)\, d\tau. \tag{A.35}$$

Let $\tau_j = j$ for $j = 0, \ldots, k-1$ and $\tau_{-1} = -1$ (the future point), and define Lagrange basis polynomials:

$$\ell_j(\tau) = \prod_{\substack{i=-1 \\ i \neq j}}^{k-1} \frac{\tau - \tau_i}{\tau_j - \tau_i}. \tag{A.36}$$

Let $P_k(\tau) = \sum_{j=-1}^{k-1} f(x - h + \tau_j h)\ell_j(\tau)$ be the Lagrange interpolant of $f(x - h + \tau h)$ using nodes $\tau_j$.

Now approximate the integral:

$$\int_0^1 \widehat{f(x - h + \tau h)}\, d\tau = \int_0^1 P_k(\tau)\, d\tau = \sum_{j=-1}^{k-1} \left( \int_0^1 \ell_j(\tau)\, d\tau \right) f(x - h + \tau_j h). \tag{A.37}$$

Define:

$$\beta_j := \int_0^1 \ell_j(\tau)\, d\tau, \quad j = -1, 0, \ldots, k-1. \tag{A.38}$$

Then we get:

$$\int_x^{\widehat{x+h}} f(s)\, ds = h \left( \sum_{j=-1}^{k-1} \beta_j f(x - h + \tau_j h) \right) = h \left( \beta_{-1} f(x - h) + \sum_{j=0}^{k-1} \beta_j f(x + jh) \right). \tag{A.39}$$

Substitute into the ODE integral:

$$\hat{y}_{n-1} = y_n + h \left( \beta_{-1} f_{n-1} + \sum_{j=0}^{k-1} \beta_j f_{n+j} \right).$$

(A.40)

which is exactly the Adams-Moulton method of order $k$.

Finally, since this is based on Lagrange interpolation over $k+1$ nodes (including the implicit node at future time), the quadrature error is $\mathcal{O}(h^{k+1})$, leading to method of order $k$. □

We now instantiate the backward Adams-Moulton method of order 3 from Theorem B.2, and quantify its effect in discrete difference form, including error propagation.

**Theorem 3.5** (Third-order backward difference via Adams-Moulton estimation). Let $x_t$ be the trajectory satisfying the ODE: $\frac{dx_t}{dt} = y_t$. Consider estimating $x_{t-1}$ by third-order backward difference $\Delta^{(3)} x_{t-1}$ using forward values $x_t, x_{t+1}, x_{t+2}$, where the step size is $\Delta t$. Furthermore, we using second and third order of Adams-Moulton method to define the estimator:

$$\hat{x}_{t-1} := x_t - \frac{5\Delta t}{6} y_t - \frac{5\Delta t}{6} y_{t+1} + \frac{2\Delta t}{3} y_{t+2}.$$

(A.41)

This estimate is consistent with the discrete ODE, and the local truncation error satisfies:

$$\hat{x}_{t-1} - x_{t-1} = \mathcal{O}(\Delta t^2).$$

(A.42)

*Proof.* By Theorem B.2, the third-order Adams-Moulton method in reverse-time (backward extrapolation) reads:

$$\widehat{x_{t-1} - x_t} = \frac{\Delta t}{12} \left( 8y_t - y_{t+1} + 5y_{t-1} \right).$$

(A.43)

In our scenario, we aim to express everything in terms of $x_t, x_{t+1}, x_{t+2}$, and $y_t, y_{t+1}, y_{t+2}$.

By using the identity:

$$\Delta^{(3)} x_{t-1} = x_{t-1} - 3x_t + 3x_{t+1} - x_{t+2},$$

(A.44)

we reverse it to get the original estimation of $\tilde{x}_{t-1}$:

$$\tilde{x}_{t-1} = 3x_t - 3x_{t+1} + x_{t+2} = x_t + 2(x_t - x_{t+1}) - (x_{t+1} - x_{t+2}).$$

(A.45)

Then we plug in a higher-order correction from the Adams-Moulton method:

$$x_t - x_{t+1} = -\frac{\Delta t}{12}(5y_t + 8y_{t+1} - y_{t+2}) + \mathcal{O}(\Delta t^3), \quad x_{t+1} - x_{t+2} = -\frac{\Delta t}{2}(y_{t+1} + y_{t+2}) + \mathcal{O}(\Delta t^2).$$

(A.46)

Combining these, we define the estimator $\hat{x}_{t-1}$:

$$\hat{x}_{t-1} = x_t - \frac{5\Delta t}{6} y_t - \frac{5\Delta t}{6} y_{t+1} + \frac{2\Delta t}{3} y_{t+2}.$$

(A.47)

Thus we have the error bound

$$
\begin{aligned}
\hat{x}_{t-1} - x_{t-1} &= x_t - \frac{5\Delta t}{6}y_t - \frac{5\Delta t}{6}y_{t+1} + \frac{2\Delta t}{3}y_{t+2} - x_{t-1} \\
&= \tilde{x}_{t-1} + \mathcal{O}(\Delta t^2) - x_{t-1} \\
&= \mathcal{O}(\Delta t^2),
\end{aligned}
\tag{A.48}
$$

as claimed.

$\square$

**Theorem 3.6** (Error bound for final reconstruction $\hat{x}_0^t$). Let $\hat{x}_0^t$ denote the reconstruction of $x_0$ at time $t$, obtained using an extrapolated estimate $\hat{x}_{t-1}$ from Theorem 3.5, which incurs an error of order $\mathcal{O}(\Delta t^2)$. Then, the final reconstruction $\hat{x}_0^t$ satisfies the following error bound:

$$
\|\hat{x}_0^t - x_0^t\| = \mathcal{O}(\Delta t) + \mathcal{O}(\Delta x_t).
\tag{A.49}
$$

*Proof.* This follows from the linearity of the prediction formula for $\hat{x}_0^t$, which is a linear combination of $x_t$ and $\hat{\epsilon}_t$. Therefore, by following **Assumption 1** that $\epsilon_\theta(x_t, t)$ is Lipschitz in $x_t$ and $t$

$$
\|\hat{x}_0^t - x_0^t\| \leq C_1\|\hat{x}_t - x_t\| + C_2\|\hat{\epsilon}_t - \epsilon_t\| = \mathcal{O}(\Delta t) + \mathcal{O}(\Delta x_t),
\tag{A.50}
$$

where $C_1$ and $C_2$ are constants depending on the schedule. $\square$

## C. Analysis on Existing Token Reduction Methods

Compared to the baseline, token merging with a high merging ratio significantly loses detailed information, while token pruning at the same ratio partially preserves fine-grained details. However, pruning still leads to the loss of a substantial amount of information in the generated image. In this section, we provide a short analysis of token merging, token pruning, and the unmerging process.

**Token merging** We represent the token-merging procedure via an $N' \times N$ matrix $M$:

$$
M_{j,i} = \begin{cases} \frac{1}{|S_j|}, & \text{if } i \in S_j, \\ 0, & \text{otherwise}, \end{cases}
$$

where each set $S_j \subseteq \{1, \ldots, N\}$ groups tokens deemed similar. The merged sequence $\mathbf{x}'$ is then obtained by:

$$
\mathbf{x}' = M\,\mathbf{x}.
$$

Within each subset $S_j$, the operation is an average: $h_j = \frac{1}{|S_j|}\mathbf{1}^T$. From a signal-processing perspective, such an averaging operation has a *low-pass* frequency response described by the sinc function:

$$
H(u) = \frac{\sin(\pi\,u\,|S_j|)}{\pi\,u\,|S_j|}.
$$

Hence, merging tokens suppresses high-frequency components while retaining lower-frequency content.

**Token pruning** Token pruning also cause information loss, as any unique feature captured by the pruned tokens are no longer available for subsequent computations. Since it removes tokens without averaging, token pruning does not exhibit the low-pass filtering effect, resulting in the loss of specific spatial information rather than a smoothing of the input sequence.

**Unmerging process** Furthermore, due to the non-linear mapping of self-attention, the high similarity of two input tokens $x[i], x[j]$ (where $i \in S_j$) does not guarantee the high similarity of corresponding outputs after attention computation. Therefore, the unmerging procedure, which duplicates the processed merged tokens back to their original position results in an inherent downsampling effect. This duplication does not recover the high-frequency details lost during merging, thus further degrading the detailed features in the output.

# D. Additional Experiments

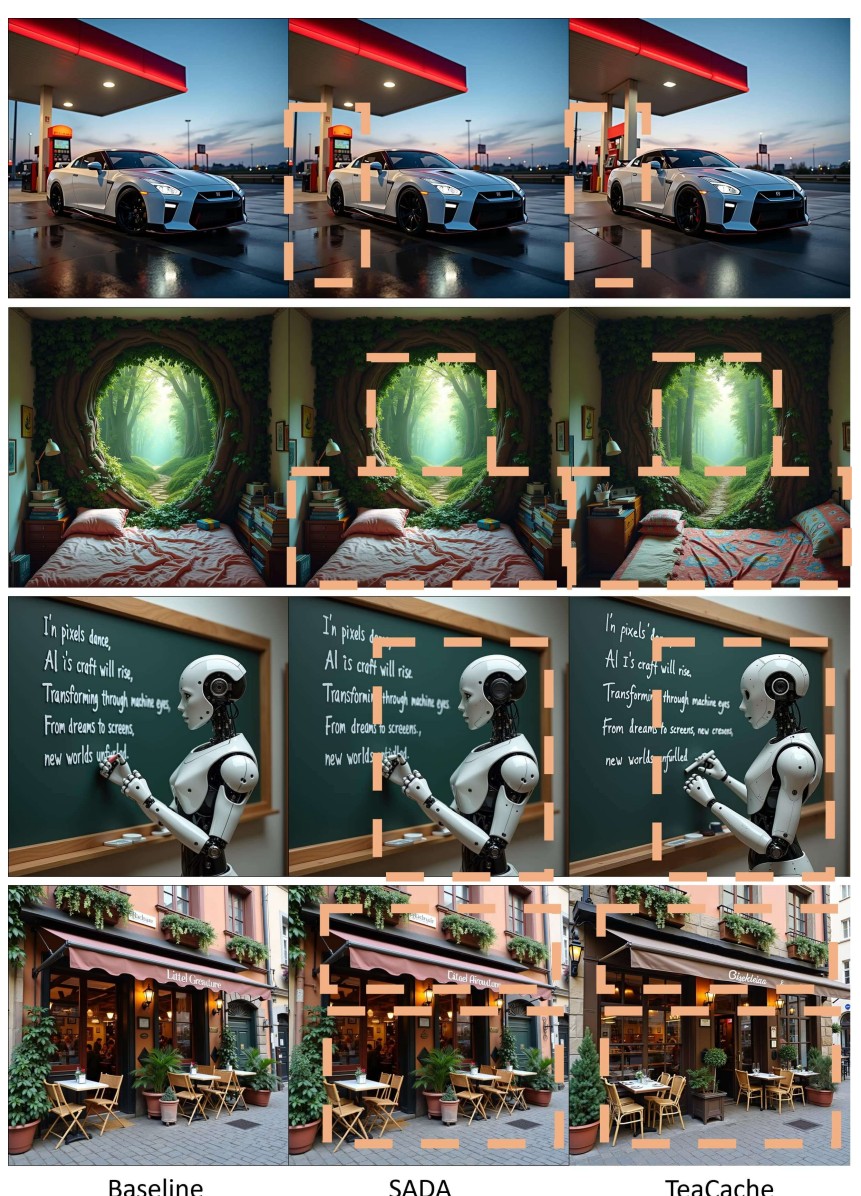

Baseline          SADA          TeaCache

*Figure A.1.* Comparison between SADA and TeaCache on FLux.1 Dev. Our method shows significantly better faithfulness under an identical speedup ratio of $2.0\times$.

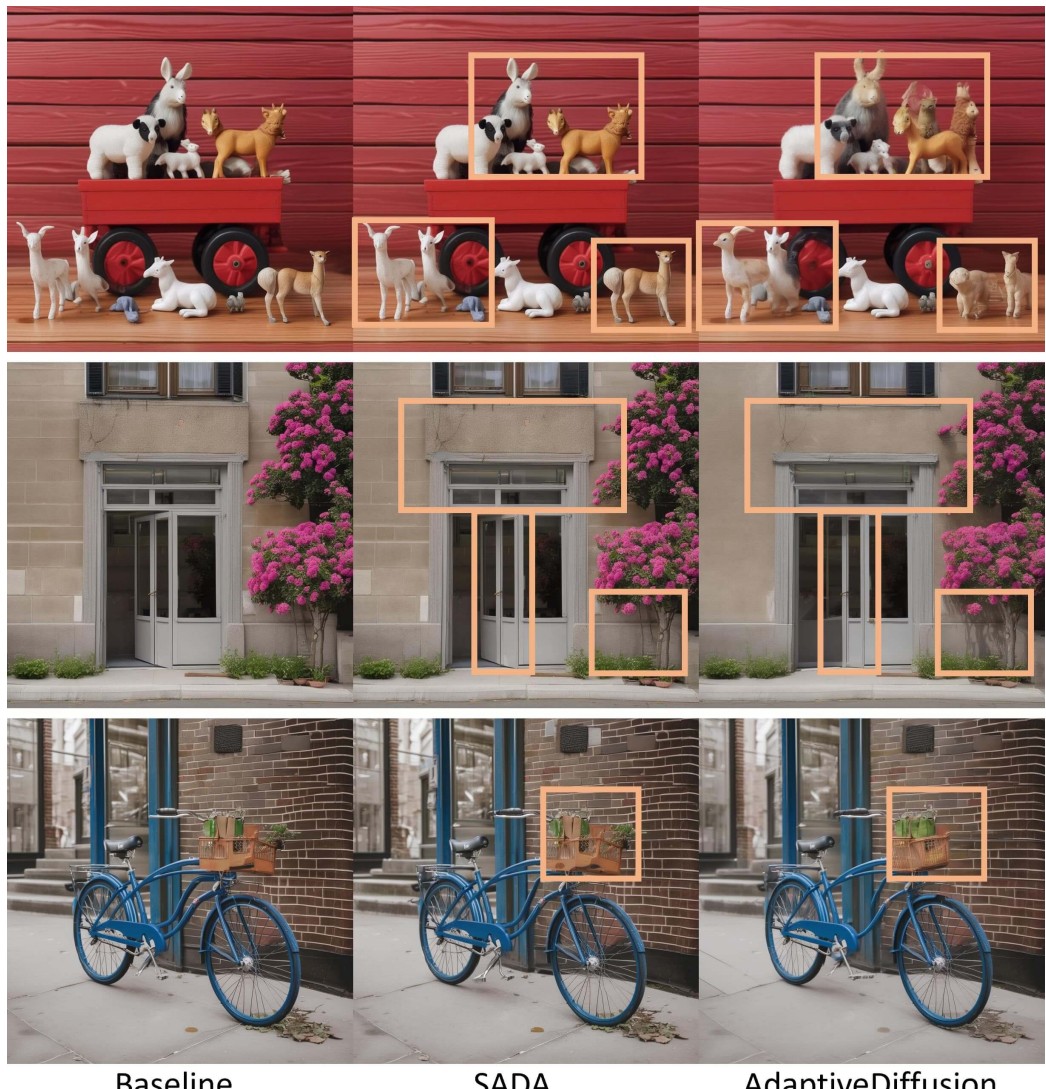

Baseline       SADA       AdaptiveDiffusion

*Figure A.2.* Comparison between SADA and AdaptiveDiffusion on SDXL with 50 steps DPM++. Our method shows better faithfulness with a much faster speedup of $1.81\times$ vs $1.65\times$.

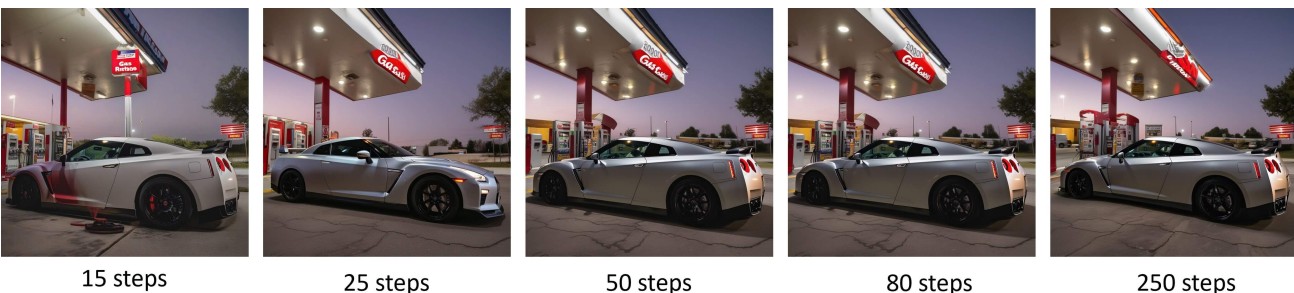

15 steps      25 steps      50 steps      80 steps      250 steps

*Figure A.3.* The generated samples from DPM-Solver first dramatically change, then demonstrate convergence after setting the sample step to 50. Therefore, we believe our baseline setting is reasonable.

