# OpenReview forum: "SADA: Stability-guided Adaptive Diffusion Acceleration"
_ICML.cc/2025/Conference — ICML 2025 poster_

### Official Review · Reviewer_4E6r · 2025-02-18

**Overall Recommendation:** 3

**Summary:**

This paper proposes SADA, a novel paradigm that unifies step-wise and tokenwise sparsity decisions using a shared criterion based on the denoised latent x0. By aligning with modern numerical solvers that rely heavily on x0, SADA offers more stable pruning decisions and preserves important visual details throughout the denoising trajectory. Extensive experiments on SD 2 and SDXL demonstrate that SADA significantly accelerates inference without compromising image quality.

**Claims And Evidence:**

Yes

**Essential References Not Discussed:**

No

**Experimental Designs Or Analyses:**

Yes

**Methods And Evaluation Criteria:**

Yes

**Other Comments Or Suggestions:**

N/A

**Other Strengths And Weaknesses:**

**Strengths:**
1. The paper is well-organized and includes comprehensive technical details.
2. The authors provide sufficient theoretical analysis for the proposed method, including the proof of error bound.
3. The images generated by the proposed method are consistent with those produced by the original diffusion models.


**Weakness:**
1. This paper only presents experimental results on U-Net-based diffusion models. Since the state-of-the-art image diffusion models now primarily use DiT or MM-DiT architectures, it is essential to demonstrate the effectiveness of SADA on models like PixArt, Flux, or SD 3.
2. The evaluation setting for DPM-Solver++ uses 50 sampling steps. However, the main advantage of DPM-Solver is its ability to achieve high-generation quality with fewer sampling steps. Therefore, it would be more reasonable to set the sampling step of DPM-Solver++ to 20.
3. The proposed SADA performs worse than AdaptiveDiffusion when evaluated with DPM-Solver++. However, DPM-Solver++ holds greater practical value compared to the Euler solver.
4. As shown in the results, the acceleration ratio of SADA is around 1.5×. Can its speedup ratio be extended to 2× or beyond at the cost of some performance?

**Questions For Authors:**

Please see the weakness

**Relation To Broader Scientific Literature:**

This paper proposed a new cache-based accelerating method in the diffusion model area.

**Theoretical Claims:**

Yes

---

> ### Author Rebuttal · Authors · 2025-04-01
>
> We sincerely thank the reviewer for the insightful and constructive comments.
>
> **Q1: Can its speedup be extended to $2 \times$ or beyond?**
>
> **A1:** Yes, it can. To apply an faster configuration, we leverage the inherent stability of the per‐step data reconstruction $x_0^t$. When the $x_0^t$ trajectory demonstrates high stability (e.g., the second half of Fig.2), we could employ larger step sizes compensated by higher-order approximations. Building on this insight, we implement a uniform step-wise pruning strategy after the stability of the denoising process with Lagrange interpolation for correction.
>
> For example, consider a 50‐step process. To achieve a step-wise pruning interval of 4 after stable (i.e., compute every 4th step fully and interpolate the skipped steps via Lagrange), we store $\hat{x}_0^t$ every 4 steps before stabilization. Their indices define the fixed-size set $I$, which is updated dynamically to limit memory usage. For any skipped $t$:
>
> $$
> \hat{x}_0^{t}\gets\sum _{i \in I}\prod _{j\in I\setminus \\{ i \\}}\frac{t-t_j}{t_i-t_j}\hat{x}_0^{t_i}
> $$
>
> Under this setting, we yield a $\geq 1.8 \times$ speedup regardless of models or solvers. The acceleration would be even more aggressive if further increasing the step size.
> To balance the degradation, we raise the Adams-Moulton approximation from second to third order, allowing $x_0^t$ to leverage information from the previous three steps (instead of two), thereby improving numerical accuracy and robustness. Our updated result in Table 1 demonstrates the effectiveness of the above improvements. Notably, we achieve a $2.02 \times$ speedup on the most powerful Flux.1 model with impressive $0.06$ LPIPS and $1.95$ FID.
>
> **Table 1: Quantitative results on MS-COCO 2017**
> |**Model**|**Scheduler**|**Methods**|**PSNR**|**LPIPS**|**FID**|**Speedup Ratio**|
> |-|-|-|-|-|-|-|
> |SD2|DPM++|DeepCache|17.70|0.271|7.83|1.43|
> |||AdaptiveDiffusion|24.30|0.100|4.35|1.45|
> |||SADA|**26.34**|**0.094**|**4.02**|**1.80**|
> |SD2|Euler|DeepCache|18.90|0.239|7.40|1.45|
> |||AdaptiveDiffusion|21.90|0.173| 7.58| **1.89**|
> |||SADA | **26.25**|**0.100**|**4.26**|1.81|
> |SDXL|DPM++| DeepCache|21.30|0.255|8.48|1.74|
> |||AdaptiveDiffusion| 26.10|0.125|4.59|1.65|
> |||SADA| **29.36**| **0.084**|**3.51**|**1.86**|
> |SDXL|Euler|DeepCache|22.00|0.223|7.36|**2.16**|
> |||AdaptiveDiffusion|24.33|0.168|6.11|2.01|
> |||SADA|**28.97**|**0.093**|**3.76**|1.85|
> |Flux|Flow-matching|TeaCache|19.14|0.216|4.89|2.00|
> |||SADA|**29.44**|**0.060**|**1.95**|**2.02**|
>
> **Q2: Ablations on DPM-Solver++**
>
> We appreciate the reviewers for highlighting the practical importance of DPM++ and its ability to achieve high quality with fewer steps.
>
> **a. SADA performs worse than AdaptiveDiffusion?**
>
> To counterbalance the aggressive configuration, we increased the order of the Adams-Moulton approximation from second to third order. This enhancement incorporates additional information from the previous denoising trajectory, which in turn improves both accuracy and stability. As shown in Table 1, SADA now significantly outperforms AdaptiveDiffusion when used with DPM++.
>
> **b. Set the sampling step to 20?**
>
> Table 2 presents a comprehensive ablation study across various sampling steps. Our method achieves a 1.5× acceleration in the 25-step scenario with negletable difference. Furthermore, we observe that as the number of inference steps increases, the images generated by DPM++ initially change dramatically before converging when the base step is set to 25. An illustration, along with additional generation examples and comparisons, is available at the following link:
>
> https://drive.google.com/file/d/168ovZu9fxcfY5PfE8F5AgkN4la6dvH9f/view?usp=sharing
>
> **Table 2: Ablation studyon sampling steps**
> |**Model**|**Scheduler**|**Steps**|**PSNR**|**LPIPS**|**FID**|**Speedup Ratio**|
> |-|-|-|-|-|-|-|
> |SD-2|DPM++|50|26.34|0.094|4.02|1.80|
> |||25|28.15|0.073|3.13|1.48|
> |||15|29.84|0.072|3.05|1.24|
> ||Euler|50|26.25|0.100|4.26|1.81|
> ||| 25| 26.83| 0.088|3.87|1.48|
> |||15|29.34|0.076|3.70|1.25|
> |SDXL|DPM++|50|29.36|0.084|3.51|1.86|
> |||25|30.84|0.073|2.80|1.52|
> |||15|31.91|0.073|2.54|1.29|
> ||Euler|50|28.97|0.093|3.76|1.85|
> |||25|29.42|0.085|3.13|1.50|
> |||15|31.28|0.084|3.26|1.26|
>
>
> **Q3. SADA for Flow-matching & DiT Architecture**
>
> Under the flow matching objective, the model directly predicts the transportation vector field $dx/dt$ between noise and data distributions. Since the denoising trajectory is ODE-based, our criterion effectively measures its stability. Table 3 on Flux (DiT) shows that our method significantly outperforms the most recent work suggested by reviewer DBkM.
>
> **Table 3: Quantitative results on MS-COCO 2017**
>
> |**Model**|**Scheduler**|**Methods**|**PSNR**|**LPIPS**| **FID**|**Speedup Ratio**|
> |--|--|--|--|--|--|--|
> | Flux|Flow-matching|TeaCache|19.14|0.216|4.89|2.00|
> |||SADA|**29.44**| **0.060**|**1.95**|**2.02**|

---

> > ### Comment · Reviewer_4E6r · 2025-04-03
> >
> > Thanks for the author's rebuttal. I raise my score to 3 weak accept.

---

> > > ### Author Response · Authors · 2025-04-03
> > >
> > > We sincerely appreciate your updated rating and positive recognition. We are genuinely pleased with your emphasis that our paper is **well-organized** and has **solid theoretical proofs**. We are particularly grateful for your support regarding our objective of training-free acceleration, which maintains **consistency with the original diffusion model**.
> > >
> > > Your insightful feedback has encouraged and inspired us to further investigate SADA's potential. We have raised the Adam-Moulton method to third-order, leading to significantly improved generative performance and efficiency compared to previous baselines. Meanwhile, we have verified the superior performance of SADA on state-of-the-art flow-matching models with MM-DiT architecture
> > >
> > > We are committed to open-sourcing the SADA plug-in package (diffuser & comfyUI), enabling training-free acceleration of existing diffusion models (and their variants) with just a single line of configuration code.
> > >
> > > Thank you again for the time and efforts put in reviewing. Should you have additional questions or suggestions, please do not hesitate to reach out.

---

### Official Review · Reviewer_km7j · 2025-03-11

**Overall Recommendation:** 5

**Summary:**

The paper proposes SADA (Stability-guided Adaptive Diffusion Acceleration), a method to accelerate diffusion models by jointly optimizing step-wise and token-wise sparsity using a unified criterion based on the denoised latent \( x_0 \). Key contributions include: (1) alignment of pruning decisions with \( x_0 \)-based solvers for stability, (2) a second-order Adams-Moulton approximation for skipped steps, and (3) a token cache mechanism to mitigate information loss. Experiments on Stable Diffusion 2 and SDXL demonstrate up to 1.52× speedup while maintaining image quality (e.g., LPIPS of 0.118 on SDXL). The method outperforms baselines like DeepCache and AdaptiveDiffusion in metrics such as FID and LPIPS.

**Claims And Evidence:**

The claims are largely supported by experiments, but some aspects need clarification:
- The assertion that \( x_0 \)-based pruning is "more stable" than \( x_t \)-based methods is validated via metrics (Table 1), but direct ablation studies comparing \( x_0 \) vs. \( x_t \) criteria are missing.

**Essential References Not Discussed:**

N/A. However, the following works could be included in the related work section or used as baselines to enhance the quality of the paper, as they also focus on training-free acceleration of diffusion models.
- Delta-DiT: A Training-Free Acceleration Method Tailored for Diffusion Transformers
- Faster Diffusion: Rethinking the Role of UNet Encoder in Diffusion Models
- Cache Me if You Can:  Accelerating Diffusion Models through Block Caching

**Experimental Designs Or Analyses:**

- Table 1 shows strong results, but baselines like ToMeSD or concurrent methods (e.g., DiT-FastAttn) are omitted.
- The ablation study (Table 2) reports improved quality with fewer steps. The authors should clarify if this stems from their method’s stability or experimental setup.

**Methods And Evaluation Criteria:**

- **Methods**: Combining step/token pruning via \( x_0 \)-alignment is novel and sensible. The Adams-Moulton approximation and token cache are well-motivated.
- **Evaluation**: COCO-2017, SD2/SDXL, and standard metrics (LPIPS, FID) are appropriate. However, user studies or qualitative examples (beyond Fig. 5) would strengthen claims about preserved visual details.

**Other Comments Or Suggestions:**

N/A

**Other Strengths And Weaknesses:**

- **Strengths**: Solid theoretical proof, Novel unification of step/token pruning, strong empirical results, and practical speedup.
- **Weaknesses**: No analysis of computational overhead from the cache mechanism.

**Questions For Authors:**

1. Few-Step Improvement: Why does LPIPS improve with fewer steps (Table 2)? Could you provide some insightful explanations?
2. Baseline Comparison: It would be better to add ToMeSD or concurrent methods such as DiT-FastAttn for comparison.
3. Some of the latest diffusion models are trained based on flow matching loss, and whether this method is also suitable for such models.
4. The diffusion model of DiT architecture has also received a lot of attention recently, whether this method is also applicable to this architecture, and if so, increasing the experimental results of this architecture will help to improve the quality of the paper.

**Relation To Broader Scientific Literature:**

The work builds on diffusion acceleration via step skipping (DPM-Solver++, AdaptiveDiffusion) and token reduction (ToMeSD, DeepCache). It unifies these paradigms, addressing limitations in prior isolated approaches. The \( x_0 \)-alignment aligns with modern ODE solvers (Karras et al., 2022), extending their utility to sparsity decisions.

**Theoretical Claims:**

Overall, the theoretical proof is solid. There are two possible concerns:
- **Theorem 3.1** (global token average): Proof in Appendix A.1 applies Lindeberg-Feller CLT but assumes independent tokens, which diffusion latents may not satisfy?
- **Theorem 3.2** (error bound): The proof assumes Lipschitz continuity of \( \epsilon_\theta \), which is standard but not empirically verified. Or it would be better to have some literature support.

---

> ### Author Rebuttal · Authors · 2025-04-01
>
> We sincerely appreciate the reviewer's thoughtful feedback and kind support for our work.
>
> Based on the suggestions from Reviewer 4E6r and DBkM, we implement an aggressive version of SADA:
>
> 1. Implementing uniform step-wise pruning when the $x_0^t$ trajectory is stable, using Lagrange interpolation.
>
> 2. Mitigating degradation by upgrading the Adam-Moulton Approximation from second to third order.
>
> Detailed motivation and formulation are provided in our response to Reviewer 4E6r, and updated results are shown in Table 1.
>
> **Q1: Comparison between $x_0$ and $x_t$-based criterion**
>
> **A1:** We compare our $x_0$ driven paradigm with AdaptiveDiffusion, which leverages the third-order difference of $x_t$ as acceleration criterion. As shown in Table 1, our method consistently delivers superior generation quality—achieving higher PSNR, lower LPIPS and FID—while maintaining a stable speed-up ratio of $\geq 1.8 \times$ regardless of model and scheduler.
>
> The $x_0$ representation is naturally aligned with the final output, capturing essential semantic structures and enabling a more robust criterion than $x_t$.
>
> **Table 1: Comparison between $x_0$ based and $x_t$ based criterion**
> |**Model**|**Scheduler**|**Methods**|**PSNR**|**LPIPS**|**FID**|**Speedup Ratio**|
> |-|-|-|-|-|-|-|
> | SD2|DPM++|AdaptiveDiffusion| 24.30|0.100| 4.35|1.45|
> |||SADA|**26.34**|**0.094**|**4.02**|**1.80**|
> |SD2|Euler|AdaptiveDiffusion|21.90|0.173|7.58|**1.89**|
> |||SADA|**26.25**|**0.100**|**4.26**|1.81|
> |SDXL|DPM++|AdaptiveDiffusion| 26.10| 0.125|4.59|1.65|
> |||SADA| **29.36**| **0.084** | **3.51**| **1.86**|
> |SDXL| Euler| AdaptiveDiffusion|24.33|0.168|6.11|**2.01**|
> |||SADA|**28.97**|**0.093**|**3.76**|1.85|
>
> **Q2 Theoretical claims**
>
> **A2-1 (CLT):** The independence assumption holds for $x_t$ as $\epsilon_t$ is sampled i.i.d. from Gaussian. For $\hat{x}_t=\sqrt{\bar{\alpha}_t}\hat{x}_0^t+\sqrt{1-\bar{\alpha}_t}\hat{\epsilon}_t$, we write $\hat{\epsilon}_t=\epsilon_t+(\hat{\epsilon}_t-\epsilon_t)$. The first term is i.i.d. Gaussian with zero mean by Law of Large Number (LLN). For the second, the training objective $E| \epsilon-\hat{\epsilon}_t|^2$ implies $\hat{\epsilon}_t\to E[\epsilon\mid x_t,t]$, so $E[\hat{\epsilon}_t-\epsilon_t]\to 0$, and by LLN, the sample mean $\overline{\hat{\epsilon}_t-\epsilon_t}\to 0$.
>
> **A2-2 (Lipschiz):** The Lipschiz continuity of $\epsilon_\theta$ is widely assumed by preliminary works such as Adaptive Diffusion and DPM-Solver.
>
> **Q3 Computational overhead**
>
> **a.** *Memory*: For step-wise pruning with third-order Adam-Moulton, after reformulate we only need to store 1 previous $x_0^t$ and 2 previous $dx/dt$ in the cache. For token-wise pruning, we store 1 previous representation $\mathbf{x}^l_t$ for only transformers with the highest resolution. For example, we observe only a neglectable increase in memory usage in the SD-XL model (from 14981 MB to 15127 MB).
>
> **b.** *Complexity*: All computation in the SADA framework is addition and scaling, $O(N)$. Note that SADA does not include any quadratic complexity computation (e.g. cosine similarity, matrix calculation) as in previous works.
>
> **Q4. Qualatative examples & Fewer step generation**
>
> We provide the following link for more generation samples and comparison with previous strategy. In Addition, the CLIP score for generation quality is provided.
>
> https://drive.google.com/file/d/168ovZu9fxcfY5PfE8F5AgkN4la6dvH9f/view?usp=sharing
>
> We appreciate for pointing out the better similarity when decreasing sampling steps. We believe the extent accumulation of error decrease when reducing sampling steps. This trend could be clearly found in our ablation table.
>
> **Q5. Comparison with other token-wise sparisty strategies**
>
> Table 2 shows that our method significantly outperforms ToMeSD. DiTFastAttention is limited to traditional Diffusion Transformer because its windowed attention cannot handle mixed-modality inputs (e.g., MM-DiT modules in SD-3 and Flux). In contrast, our approach easily adapts to these architectures, as demonstrated later.
>
> **Table 2: Quantitative results on MS-COCO 2017**
>
> |**Model**|**Scheduler**|**Methods**|**PSNR**|**LPIPS**| **FID**|**Speedup Ratio**|
> |-|-|-|-|-|-|-|
> |SD2|DPM++|ToMeSD|16.29|0.41|13.70|1.10|
> |||SADA|**26.34**|**0.094**|**4.02**|**1.80**|
>
> **Q6. SADA for Flow-matching & DiT Architecture**
>
> Under the flow matching objective, the model directly predicts the transportation vector field $dx/dt$ between noise and data distributions. Since the denoising trajectory is ODE-based, our criterion effectively measures its stability. Table 3 on Flux (DiT) shows that our method significantly outperforms the most recent work suggested by reviewer DBkM.
>
> **Table 3: Quantitative results on MS-COCO 2017**
>
> |**Model**|**Scheduler**|**Methods**|**PSNR**|**LPIPS**| **FID**|**Speedup Ratio**|
> |-|-|-|-|-|-|-|
> | Flux|Flow-matching|TeaCache|19.14|0.216|4.89|2.00|
> |||SADA|**29.44**| **0.060**|**1.95**|**2.02**|

---

> > ### Comment · Reviewer_km7j · 2025-04-02
> >
> > Thank you for your response. My concerns have been mostly addressed. The adaptive mechanism in diffusion models has rarely been studied before and holds great significance; therefore, I consider this work a valuable contribution to the diffusion model community. The additional experimental results provided in the rebuttal further validate the effectiveness of the proposed method. As a result, I am inclined to raise my score.

---

> > > ### Author Response · Authors · 2025-04-02
> > >
> > > We sincerely thank you for your thoughtful review and support. We deeply appreciate your recognition of the **novelty and great significance of adaptive mechanisms in diffusion models**, **the uniqueness of dynamic allocation of token-wise and step-wise sparsity**, and **the solid theoretical proof** — which makes our proposed approach accelerate generative modeling by dynamically adjusting configurations for different prompts while best preserving faithfulness.
> > >
> > > We are delighted that our additional experimental results and analysis have addressed your concerns. Your constructive feedback is very important for us to refine and improve our approach.
> > >
> > > We look forward to releasing the SADA package to the diffusion community in the camera-ready phase!

---

### Official Review · Reviewer_DBkM · 2025-03-12

**Overall Recommendation:** 1

**Summary:**

The paper proposes SADA, a training-free acceleration method for diffusion models that unifies step-wise (temporal) and token-wise (spatial) sparsity using a stability criterion based on the denoised latent $ x_0 $. Specifically,  the paper uses a unified $ x_0 $-guided sparsity criterion for step skipping and token pruning, leveraging $ x_0 $'s structural alignment with modern ODE solvers.  A second-order Adams-Moulton method to approximate skipped steps and a token cache to reconstruct pruned tokens.  Experiments on Stable Diffusion 2 and XL show speedups of up to 1.5× while maintaining image quality.

## update after rebuttal
Thank you to the authors for their response and additional experiments, which have provided me with a deeper understanding of SADA's effectiveness. However, the baselines compared in this paper are somewhat non-repetitive, and some important papers on cache-based DiT acceleration, such as Learning-to-Cache and $\Delta - Dit$, were not included in the comparison. Additionally, my concerns about the novelty of this paper remain. The caching method proposed in the paper does not differ fundamentally from previous approaches. Although the work most similar to this paper, [1], was published during the review period, earlier works like [2, 3, 4] also bear significant similarity in methodology, especially TeaCache [4]. A comprehensive experimental comparison with these papers, along with a detailed explanation of the differences in approach, is necessary. I still recommend rejecting this paper.

[1] Token-aware and step-aware acceleration for stable diffusion

[2] Cached Adaptive Token Merging: Dynamic Token Reduction and Redundant Computation Elimination in Diffusion Mode

[3] Accelerating diffusion transformers with token-wise feature caching

[4] Timestep Embedding Tells: It's Time to Cache for Video Diffusion Model

**Claims And Evidence:**

- **Claim 1**: $ x_0 $-based pruning improves stability and aligns with solvers.

    *Evidence*: Theoretical analysis (Theorem 3.1) links $ x_0 $ to step stability; experiments show lower LPIPS/FID than $ x_t $-based methods (Table 1).

    *Problems*: Limited comparison to other $ x_0 $-aligned methods; no ablation on $ x_0 $ vs. $ x_t $.

- **Claim 2**: Unified sparsity outperforms isolated strategies.

    *Evidence*: SADA outperforms DeepCache/AdaptiveDiffusion in FID/LPIPS (Table 1).

    *Problems*: Missing comparisons to recent works[1,2,3,4]

[1] Token-aware and step-aware acceleration for stable diffusion

[2] Cached Adaptive Token Merging: Dynamic Token Reduction and Redundant Computation Elimination in Diffusion Mode

[3] Accelerating diffusion transformers with token-wise feature caching

[4] Timestep Embedding Tells: It's Time to Cache for Video Diffusion Model

**Essential References Not Discussed:**

The following references should be discussed because they are similar in method to this paper.

[1] Token-aware and step-aware acceleration for stable diffusion

[2] Cached Adaptive Token Merging: Dynamic Token Reduction and Redundant Computation Elimination in Diffusion Mode

[3] Accelerating diffusion transformers with token-wise feature caching

[4] Timestep Embedding Tells: It's Time to Cache for Video Diffusion Model

**Experimental Designs Or Analyses:**

- **Strengths**: Broad evaluation across schedulers (DPM++, Euler) and models (SD2, SDXL).
- **Weaknesses**:
   + No analysis of computational overhead from the token cache or varying pruning ratios.
   + No clipscore or pick-socre reported, The former evaluated the text alignment of images generated by models after acceleration, while the latter assessed aesthetic scores.
   + Missing comparisons to recent works[1,2,3,4]
   + The acceleration effect obtained is not significant compared to previous work; it's around 1.5x, which is incremental in nature. The ideas presented in this article do not differ significantly from those based on cache methods previously [1,2,3,4], with contributions being incremental as well.

[1] Token-aware and step-aware acceleration for stable diffusion

[2] Cached Adaptive Token Merging: Dynamic Token Reduction and Redundant Computation Elimination in Diffusion Mode

[3] Accelerating diffusion transformers with token-wise feature caching

[4] Timestep Embedding Tells: It's Time to Cache for Video Diffusion Model

**Methods And Evaluation Criteria:**

- **Methods**: The $ x_0 $-aligned criterion and reconstruction mechanisms are well-motivated. Adams-Moulton provides a principled ODE-based approximation.
- **Evaluation**: COCO-2017 benchmarks with standard metrics (LPIPS, FID) are appropriate. But no clipscore or pick-socre reported, The former evaluated the text alignment of images generated by models after acceleration, while the latter assessed aesthetic scores.

**Other Comments Or Suggestions:**

This paper is poorly written and difficult to read.

**Other Strengths And Weaknesses:**

no

**Questions For Authors:**

see Weaknesses

**Relation To Broader Scientific Literature:**

Aligns with ODE-based solvers (DPM-Solver++) and token reduction (ToMe). Missing discussion of previous similar work[1,2,3,4]

[1] Token-aware and step-aware acceleration for stable diffusion

[2] Cached Adaptive Token Merging: Dynamic Token Reduction and Redundant Computation Elimination in Diffusion Mode

[3] Accelerating diffusion transformers with token-wise feature caching

[4] Timestep Embedding Tells: It's Time to Cache for Video Diffusion Model

**Theoretical Claims:**

- **Theorem 3.1** (Lindeberg condition): Correct under the assumption of token independence, but real-world spatial correlations in images may affect validity.
- **Theorem 3.2** (Error bound): Relies on Lipschitz continuity of $ \epsilon_\theta $, which is not empirically validated.

---

> ### Author Rebuttal · Authors · 2025-04-01
>
> We thank the reviewer for comprehensive comments.
>
> **Q1: Aggressive configuration of SADA**
>
> **A1:** To implement an aggressive version, we leverage the inherent robustness of the per‐step data reconstruction $x_0^t$. When the $x_0^t$ trajectory demonstrates high stability (e.g., the second half of Fig.2), we could employ larger step sizes compensated by higher-order approximations. Building on this insight, we implement a uniform step-wise pruning strategy after the stability of the denoising process with Lagrange interpolation for correction.
>
> For example, consider a 50‐step process. To achieve a step-wise pruning interval of 4 after stable (i.e., compute every 4th step fully and interpolate the skipped steps via Lagrange), we store $\hat{x}_0^t$ every 4 steps before stabilization. Their indices define the fixed-size set $I$, updated dynamically to limit memory usage. For any skipped $t$:
>
> $$
> \hat{x}_0^{t}\gets\sum _{i \in I}\prod _{j\in I\setminus \\{ i \\}}\frac{t-t_j}{t_i-t_j}\hat{x}_0^{t_i}
> $$
>
> Under this setting, we yield a $\geq 1.8\times$ speedup regardless of models or solvers. To balance the degradation, we raise the Adams-Moulton approximation from second to third order, allowing $x_0^t$ to leverage information from the previous three steps (instead of two), thereby improving numerical accuracy and robustness. Our updated result in Table 1 demonstrates the effectiveness of the above improvements.
>
> **Table 1: Quantitative results on MS-COCO 2017**
> |**Model**|**Scheduler**|**Methods**|**PSNR**|**LPIPS**|**FID**|**Speedup Ratio**|
> |-|-|-|-|-|-|-|
> |SD2|DPM++|DeepCache|17.70|0.271|7.83|1.43|
> |||AdaptiveDiffusion|24.30|0.100|4.35|1.45|
> |||SADA|**26.34**|**0.094**|**4.02**|**1.80**|
> |SD2|Euler|DeepCache|18.90|0.239|7.40|1.45|
> |||AdaptiveDiffusion|21.90|0.173| 7.58|**1.89**|
> |||SADA | **26.25**|**0.100**|**4.26**|1.81|
> |SDXL|DPM++| DeepCache|21.30|0.255|8.48|1.74|
> |||AdaptiveDiffusion| 26.10|0.125|4.59|1.65|
> |||SADA| **29.36**| **0.084**|**3.51**|**1.86**|
> |SDXL|Euler|DeepCache|22.00|0.223|7.36|**2.16**|
> |||AdaptiveDiffusion|24.33|0.168|6.11|2.01|
> |||SADA|**28.97**|**0.093**|**3.76**|1.85|
>
> **Q2: Comparison between $x_0$ and $x_t$-based criterion**
>
> **A2:** To our best knowledge, we are the first work that considers $x_0$ as acceleration criterion and approximation objective. We compare our paradigm with AdaptiveDiffusion, which leverages the third-order difference of $x_t$, as demonstrated in Table 1.
>
> Residing in an image representation space, $x_0$ demonstrates structural alignment with the final output while evolving in a more robust trajectory (as shown in Fig. 2). It captures semantics and thus yields a more consistent sparsity allocation decision.
>
> **Q3: Comparisons to recent works**
>
> **A3:** We thank the reviewers for listing the four recent works, and we will cite and discuss them in the camera-ready version. The four works explore different Caching mechanisms within diffusion architectures that accelerate sampling. However, we believe our work **fundamentally differs** from the four works listed. Note that [1] is published after submission, which is impossible be addressed at the moment.
>
> **a.** *Methodology*: The four works mentioned above accelerate diffusion in a fixed configuration (e.g., fixed caching interval and pruning ratio), while SADA is adaptive to different prompts. In addition, to our best knowledge, SADA is the first work that unifies token- and step-wise sparsity by a single criterion from the perspective of the ODE-solver process, achieving a multi-granularity adaptive acceleration strategy. The novelty is strongly supported by Reviewer km7j.
>
> **b.** *Motivation*: SADA formulates the acceleration of the ODE-based generative modeling (e.g., Diffusion, Flow-matching) as a **stability measure** of the denoising trajectory, while the four works focus only on the redundancy of the denoising architecture with relatively weak theoretical justification.
>
> **c.** *Experiment*: We believe the objective of post-training acceleration is to preserve the similarity (faithfulness) between original generated and accelerated samples while maximizing speed. Therefore, we evaluate using LPIPS and FID computed between these samples—unlike previous work, which only compares the FID of accelerated samples against the dataset. As shown in Table 2 of our response to Reviewer km7j, our method significantly outperforms [4] on FLUX.1 in terms of faithfulness at the same speed-up ratio.
>
> **Q4: CLIP/Pick Score**
>
> **A4:** Our objective is to preserve the original generation quality through our sparsity framework—metrics like these do not reflect that goal. For completeness, we have provided the requested metrics, along with generation samples and comparisons, via the link below:
>
> https://drive.google.com/file/d/168ovZu9fxcfY5PfE8F5AgkN4la6dvH9f/view?usp=sharing
>
> **Q5: Computational overhead analysis & validation of Theoretical claim**
>
> Please refer to Q2, Q3 in our response to Reviewer km7j.

---

### Decision · Program_Chairs · 2025-05-01

**Decision:**

Accept (poster)

**Comment:**

The paper introduces SADA (Stability-guided Adaptive Diffusion Acceleration), a method designed to accelerate diffusion models by jointly optimizing step-wise and token-wise sparsity through a unified criterion based on the denoised latent variable $x_0$. SADA is evaluated on Stable Diffusion 2 and SDXL, demonstrating up to a 1.5× speedup while preserving image quality.

The review scores are 1–3–5. The main concerns raised by Reviewer DBkM relate to the similarity of the proposed approach to existing work (see references [1]–[4] in the corresponding review). Although these references were unpublished or unavailable at the time of the ICML submission deadline, the authors should address and discuss them in the updated manuscript.

Reviewer 4E6r’s concerns appear to have been addressed. We encourage the authors to reflect these updates in the revised version.
Additionally, the reviewers suggest further improvements, and we recommend that the authors incorporate relevant points from their rebuttal to strengthen the final version of the paper.